# Impacts of the Manaus pollution plume on the microphysical properties of Amazonian warm-phase clouds in the wet season

Micael A. Cecchini[1], Luiz A.T. Machado[1], Jennifer M. Comstock[2], Fan Mei[2], Jian Wang[3], Jiwen Fan[2], Jason M. Tomlinson[2], Beat Schmid[2], Rachel Albrecht[5], Scot T. Martin[4] and Paulo Artaxo[5]

[1]Center for Weather Forecasting and Climate Research (CPTEC), National Institute for Space Research (INPE), Brazil.
[2]Pacific Northwest National Laboratory
[3]Brookhaven National Laboratory
[4]Harvard University
[5]São Paulo University

*Correspondence to*: M. A. Cecchini (micael.cecchini@cptec.inpe.br)

**Abstract.** The remote atmosphere over the Amazon can be similar to oceanic regions in terms of aerosol conditions and cloud type formations. This is especially true during the wet season. The main aerosol-related disturbances over the Amazon have both natural sources, such as dust transport from Africa, and anthropogenic sources, such as biomass burning or urban pollution. The present work considers the impacts of the latter on the microphysical properties of warm-phase clouds by analyzing observations of the interactions between Manaus pollution plume and its surroundings, as part of the GoAmazon2014/5 Experiment. The analyzed period corresponds to the wet season (specifically from Feb to Mar 2014 and corresponding to the first Intensive Operating Period (IOP1) of GoAmazon2014/5). The droplet size distributions reported are in the range 1 μm ≤ D ≤ 50 μm in order to capture the processes leading up to the precipitation formation. The wet season largely presents a clean background atmosphere characterized by frequent rain showers. As such, the contrast between background clouds compared to those affected by the Manaus pollution can be observed and detailed. The focus is on the characteristics of the initial microphysical properties in cumulus clouds predominantly at their early stages. The pollution-affected clouds are found to have smaller effective diameters and higher droplet number concentrations. The differences range from 10% to 40% for the effective diameter and are as high as 1000% for droplet concentration for the same vertical levels. The growth rates of droplets with altitude are slower for pollution-affected clouds (2.90 compared to 5.59 μm km$^{-1}$), as explained by the absence of bigger droplets at the onset of cloud development. Clouds under background conditions have higher concentrations of larger droplets (> 20 μm) near the cloud base, which would contribute significantly to the growth rates through the collision-coalescence process. The overall shape of the droplet size distribution (DSD) does not appear to be predominantly determined by updraft strength, especially beyond the 20 μm range. The aerosol conditions play a major role in that case. However, the updrafts modulate the DSD concentrations and are responsible for the vertical transport of water in the cloud. The larger droplets found in background clouds are associated with weak water vapour competition and a bimodal distribution of droplets sizes in the lower levels of the cloud, which enables an earlier initiation of the collision-coalescence process. This study shows that the pollution produced by Manaus affects significantly warm-phase

microphysical properties of the surrounding clouds by changing the initial DSD formation. The corresponding effects on ice-phase processes and precipitation formation will be the focus of future endeavors.

## 1 Introduction

The natural atmosphere of the Amazon is a system where the forest itself provides the nuclei for clouds, which in turn
activate the hydrological cycle and help distribute the water that maintains the local flora. Under undisturbed conditions the aerosol particles that serve as cloud condensation nuclei (CCN) are mainly secondarily generated from the oxidation of biogenic gases (Pöschl et al., 2010). Primary aerosols emitted directly from the forest may also contribute to the overall CCN population and are especially active as ice nuclei (IN). A review of the cloud-active aerosol properties and sources in general is provided by Andreae and Rosenfeld (2008) and specifically for the Amazon by Martin et al. (2010). The results presented
herein relate to the local wet season, which presents a relatively clean atmosphere compared to the local dry season when biomass burning is more frequent (Artaxo et al., 2002).

Given such an environment it is interesting to study the impacts that a city like Manaus has on the atmospheric conditions. Manaus is located in the Brazilian Amazonas state, in the middle of the forest, and has a population of about 2 million people. The human activities associated with the city produce air pollution, which interacts with the natural background
gases and particles. Several studies found that city pollution enhanced atmospheric oxidation (Logan et al., 1981; Thompson, 1992; Kanakidou et al., 2000; Lelieveld et al., 2008), which not only impacts human health but also may interact with biogenic gases to increase secondary aerosol formation. Another example is the interaction between volatile organic compounds (VOCs) with the urban $NO_x$, which leads to enhanced ozone concentrations through a photochemical process (Trainer et al 1987, Chameides et al., 1992; Biesenthal et al. 1997; Starn et al. 1998; Roberts et al. 1998; Wiedinmyer et al.,
2001).

The effects that the Manaus city has on the chemical properties of the local atmosphere potentially alter the way in which clouds are formed. Not only can the human activities change particles chemical properties, they can also increase the number concentration available for droplet formation. Most of this additional particulate matter is tied to emissions from traffic and power plants in the case of Manaus. Previous studies regarding the effects of anthropogenic aerosols on Amazonian cloud
generally focused on biomass-burning related occasions (e.g. Roberts et al., 2003; Andreae et al., 2004; Freud et al., 2008, Martins and Silva Dias, 2009) in the dry or transition seasons. However, no study evaluated the urban aerosol interaction with clouds over the rain forest during the wet season, when biomass-burning is strongly reduced given the frequent rain showers that leave the forest wet and more difficult to burn. In this case, the effects of the Manaus plume can be studied separately and in detail. Polluted clouds over the Amazon usually present more numerous but smaller droplets that grow
inefficiently by collision-coalescence and therefore delay the onset of precipitation to higher altitudes within clouds (Rosenfeld et al., 2008).

The results presented herein are based on data sets collected between February and March 2014 during the first Intensive Operations Period (IOP1) of The Observations and Modeling of the Green Ocean Amazon (GoAmazon2014/5) experiment (Martin et al., submitted). The period is in the wet season, which presents a clean atmosphere due to the reduction in biomass burning. The pristine characteristic of the background air provides the opportunity for contrasting the microphysics of natural and urban pollution-affected clouds. Due to the proximity to the Intertropical Convergence Zone (ITCZ) and the trade winds, the large-scale motions are rather stable over the region for the campaign period. Most of the time, trade winds from the northeast prevail, advecting the pollution plume southwestward. This scenario allows for the first time the direct comparison between clouds formed under background conditions and those affected by pollution in the wet season.

Clouds in the wet season differ from those in the dry and transition periods both because of aerosol conditions and large-scale meteorology (Machado et al, 2004). Although there is not a complete reversal of the mean wind directions intra-annually, the wet season clouds can be related to a monsoon system, usually referred as South American Monsoon System (SAMS). Zhou and Lau (1998) suggest that the monsoon-like flow can be understood when analyzing monthly anomalies on the wind fields. During the austral summer months, the winds tend to have a stronger northeastern component over Manaus area, while at austral winter time the stronger wind component is from the southeast. More details on the SAMS, including comparisons with other monsoon systems, can be found in Vera et al. (2006).

The main objective of this work is to understand the effects that anthropogenic urban pollution have on cloud droplets properties and development in the Amazon during the wet season. Specifically, the focus is on the comparison between warm-phase properties of clouds affected and not affected by the pollution emitted from Manaus city. The urban aerosol effect will be analyzed as function of height above the cloud base and vertical velocity. Section 2 describes the instrumental setup and the methods used for the analysis. The main findings are detailed in Section 3, while the summary and discussion are presented in Section 4.

## 2 Methodology

Sixteen research flights took place near Manaus in the Amazon forest between February and March 2014. Manaus coordinates are 3º06'S, 60º01'W and the dates and time periods of the flights are listed in Table 1, with times in UTC (local time is UTC-4). The U.S. Department of Energy Atmospheric Radiation Measurement program Gulfstream-1 (G-1) airplane (Schmid et al., 2014) performed 16 flights while measuring aerosol concentrations and composition, radiation quantities, gas-phase chemistry and cloud microphysical properties. The G-1 aircraft performed mostly short-ranged flights from Manaus, with most of the observations being held closer than 100 km from Manaus. The flight patterns were mainly focused on measuring properties in and around the city pollution plume. A schematic for the concepts of the flight planning is shown in Figure 1. The actual patterns varied daily depending on the weather forecast and plume dispersion prediction (Figure 2). Additionally, other patterns were performed such as a run upwind from Manaus in order to probe a background air reference,

or cloud profiling missions (vertical slices of the cloud field). However, the kind of pattern shown in Figure 1 was the most used and is the determinant to assess the interaction between the urban plume with the background atmosphere.

During the wet season it is very common to observe cumulus clouds as exemplified in Figure 1 and the G-1 cloud measurements consisted mostly of quick penetrations in those types of systems. From Manaus airport, the aircraft performed several legs perpendicular (or as close to as possible) to the plume direction while moving away from the city. At the end of the pattern, the aircraft started over in a different altitude and performed same flight legs. In this way, it was possible to collect not only data regarding the plume but also on the surrounding background air. During the local wet season, the background atmosphere is rather clean and the effects of the plume can be readily observed. The pollution-aerosols in this situation are almost only urban and biomass-burning contribution is very exceptional. The main idea to compare the background and polluted clouds is to accumulate statistics inside and outside the plume sections as shown in Figure 1. By concatenating the observations for the different flights, it was possible to obtain a dataset of background and polluted droplet size distributions (DSDs), which can then be used to look at aerosol impacts in different ways. All G-1 flights were used in order to obtain the highest sample size possible. Figure 2 shows the trajectories for all flights, where the dashed grey lines represent the plume angular section considered from the airplane data. Note that the plume usually disperses from Manaus to the T3 site, with relatively small variations on the direction based on the wind field. Two flights (4 and 6) had low sampling on the plume given the trajectories and the grey lines may not represent the overall region of the plume. However, the directions identified presented higher CN concentrations than the other ones.

## 2.1 Instrumentation

The two main instruments used for this study were the Condensational Particle Counter (CPC, TSI model 3025), and the Fast Cloud Droplet Probe (SPEC Inc., FCDP). The CPC instrument measures number concentration of aerosols between 3 nm and 3 $\mu$m using an optical detector after a supersaturated vapor condenses onto the particles, growing them into larger droplets. Particle concentrations can be detected between 0 and $10^5$ cm$^{-3}$, with an accuracy of $\pm10\%$. Coincidence is less than 2% at $10^4$ cm$^{-3}$ concentration, and corrections are automatically applied for concentrations between $10^4$ cm$^{-3}$ and $10^5$ cm$^{-3}$. The CPC was mounted in a rack inside the cabin and connected to an isokinetic inlet and an aerosol flow diluter and was operated using an external pump. The isokinetic inlet has an up limit of 5 $\mu$m for particle diameter, with penetration efficiency higher than 96%. A 1.5 LPM flow rate was maintained using a critical orifice. The dilution factor varied between 1 and 5.

The FCDP measures particle size and concentration by using focused laser light that scatters off particles into collection lens optics and is split and redirected toward 2 detectors. The FCDP bins particles into twenty bins ranging between 1 and 50 $\mu$m, with an accuracy of approximately 3 $\mu$m in the diameters. Bin sizes were calibrated using glass beads at several sizes in the total range. The FCDP was mounted on the right wing of the G-1 aircraft. Shattering effects were filtered from the FCDP-measured DSDs (Droplet Size Distributions), which is a built-in feature of the provider software. Additionally, measurements with low number concentrations ($< 0.3$ cm$^{-3}$) and low water contents ($< 0.02$ gm$^{-1}$) were excluded.

The quality flag of the CPC instrument was used to correct the concentration measurements. Whenever an observation was flagged as "bad", it was substituted by an interpolation between the closest measurements before and after it that were either "questionable" or "good". For "good" measurements, which represents 59% of all the measurements, the uncertainty is less than 10%. The interpolation weights decayed exponentially with the time difference between the current observation and the

reference ones. If the reference observations were more than 10 s apart, these data were excluded. 16% of the data was interpolated in that manner, while only 0.02% had to be excluded. This process was required not only to smooth out the bad measurements but also was important to maintain significant sample sizes (instead of simply excluding "bad" measurements). No averaging was applied to the 1 Hz CPC data. However, tests were made in order to check the impact that the sample frequency had on the results. The results were not sensible to moving averages of up to 10 seconds, which

corresponds to roughly 1 km displacement given that the G-1 flew around 100 ms$^{-1}$ in speed. Given this observation, the analyzes are based on the 1 Hz CPC measurements.

Complementary measurements of meteorological conditions were obtained from the Aventech Research Inc. AIMMS-20 instrument (Aircraft-Integrated Meteorological Measurement System – Beswick et al., 2008). This instrument combines temperature, humidity, pressure, and aircraft-relative flow sensors in order to provide the atmospheric conditions during the

measurements. From the aircraft measurements of relative flow, the vertical wind speed was obtained and was used herein to compare cloud properties in the up and downdraft regions. The precision of vertical wind speeds is 0.75 m s$^{-1}$ at 75 m s$^{-1}$ true airspeed.

## 2.2 Plume classification

In order to compare two different populations of clouds, namely those formed under background conditions compared to

those affected by pollution, a classification scheme was developed. The most discernible and readily observable difference between a polluted and background atmosphere is the number concentration of aerosol particles per unit volume. Urban activities such as traffic emit large quantities of particles to the atmosphere, which are then transported by atmospheric motions and can participate in cloud formation, especially when they grow, age and become more effective droplet activators. Their number concentration and sizes primarily determine their role on the initial condensational growth of cloud

droplets through the aerosol activation mechanism. Even though the urban aerosols have a lower efficiency to become CCN (cloud condensation nuclei), their number concentrations are high enough to potentially produce a higher number of cloud droplets (see, for example, Kuhn et al., 2010). By affecting the initial formation of the droplets, increased aerosol concentrations due to urban activities can alter the cloud microphysical properties throughout its whole life cycle. It will be considered here that a simple, yet effective, classification scheme should consider primarily aerosol number concentrations

to discriminate polluted and background conditions with respect to cloud formation environments. The intent of the classification scheme is not to quantify specifically the aerosols concentrations available for cloud formation under background and polluted conditions. Rather, it is a way to identify atmospheric sections that presented urban or natural aerosol characteristics.

Aerosol particle number concentrations (CN) measured by the CPC-3025 instrument were used to identify the plume location. The first procedure required is the elimination of possible artifacts related to measurements while the aircraft was inside a cloud. For that purpose, a cloud mask must be considered. The data are considered to be in-cloud by examining particle concentrations detected by several aircraft probes. The aircraft probes used to determine the presence of cloud are the Passive Cavity Aerosol Spectrometer (PCASP, SPEC Inc.), the 2D-Stero Probe (2D-S), and the Cloud Droplet Probe (CDP, Droplet Measurement Technologies). The thresholds for detection of cloud are when either the PCASP bins larger than 2.8 μm have a total concentration larger than 80 cm$^{-3}$, the 2D-S total concentration is larger than 0.05 cm$^{-3}$, or the CDP total concentration is larger than 0.3 cm$^{-3}$. Thresholds were determined by examining the sensitivity of each instrument. Assuming that the presence of clouds can affect the CN measurements, the concentrations inside clouds were related to those in clear air. Whenever an in-cloud observation is detected, the CN concentration is substituted by the closest cloud-free measurement (given that they are not more than 15 s apart, in which case the data are excluded from the analysis). In this way, possible cloud and rain effects on aerosols concentrations, such as rainout or washout, can be mitigated on the analysis. A simple and fixed threshold to separate the background and polluted observations is not enough because the altitude of the measurements should also be taken into account. For that purpose, all CPC data were used to compute vertical profiles of particle number concentrations in 800-m altitude bins. This resolution was chosen in order to result in significant amounts of data in each vertical bin. A background volume is identified whenever the measured particle concentration is below the 25% quartile profile. The polluted ones are considered to be the ones above the 90% profile. Additionally, it is required that the measurement is located in the general direction of the urban pollution dispersion in order to be considered a plume volume. Similarly, the background measurements are limited to the section outside the plume location only. It is important to note that, while the CPC data are available for the whole duration of the flights, in-cloud observations are limited to the times of actual penetrations. The choice of asymmetric 25% and 90% profiles result in similar sample sizes for the classified polluted and background in-clouds measurements (305 and 424 s, respectively), while maximizing the differences between the populations.

Given the daily variations of meteorological characteristics, the plume direction, width, and overall particle concentrations may vary. For that reason, the plume angular section must be obtained for each day individually. Figure 3 shows an example of plume classification for the flight on 10 March 2014. The CN concentrations are shown as a function of the azimuth angle with respect to Manaus airport (0° is east, grows counterclockwise), irrespective of altitude. The color represents the horizontal distance (km) from the airport. Note that there is an angular section where the concentrations are high not only close to the city but also as far as 70 km. This section is defined to be affected by Manaus pollution plume (delimited by grey dashed lines in Figure 3). Note that the coordinate system is centered on Manaus' airport, where the G-1 took off, and not on the center of the city or other point of interest. For this reason, it is also possible to observe relatively high CN concentrations close to the origin and to the northeast and southeast directions. This corresponds to high CN concentrations over the city. By keeping those directions outside the plume angular section, this data is not considered as plume. This is intentional because other aspects occur over the city that may contribute to the cloud formation. For instance, the heat island effect may

contribute to the convection, changing the thermodynamic conditions compared to those over the forest. By keeping the origin point as the airport, which is located on the west section of the city, this problem is avoided.

The final result of the classification scheme for March 10 is shown in Figure 4. A visual inspection of radiosonde (released from the Ponta Pelada airport located on southern Manaus) trajectory plots confirmed the overall direction of the plume for each flight. Given the nature of the meteorology in the Amazonian wet season, i.e. its similarities with oceanic conditions concerning horizontal homogeneity, there should be no significant difference between the thermodynamic conditions inside and outside the plume region for the G-1 flights. In this way, differences observed in pollution-affected clouds are primarily due to the urban aerosol effects. It should be noted that even though the plume classification is defined from the CN measurements, there are also observable differences regarding CCN concentrations. The in-plume CCN concentrations (for altitudes lower than 1000 m) averages at 257 $cm^{-3}$ for a 0.23% supersaturation, while the respective background concentration is 107 $cm^{-3}$ (Figure 5). Note the overall low concentrations representative of the wet season. In that case, the plume increases the CCN concentrations by more than a factor of 2. For higher supersaturation conditions (which can be achieved in strong updrafts), the differences are even more pronounced. At 0.5% supersaturation, the average CCN concentration inside the plume is 564 $cm^{-3}$, while outside it is 148 $cm^{-3}$. This shows that the plume increases the concentration of aerosol particles that are able to form cloud droplets under reasonable supersaturation conditions, even though they are less efficient than the particles in the background air.

In addition to the plume, the river breeze also plays a role on the convection characteristics over the region and the respective microphysics. The clouds directly above the rivers are usually suppressed given the subsidence from the breeze circulation. This was addressed by comparing the DSDs under plume and background conditions only for measurements over land and it showed a similar picture to what will be shown in the next section. In this way it is possible to confirm that the results presented here reflects the effect of Manaus pollution plume and not the river breeze, even though the clouds over land were indeed more vigorous. The results shown on the next section consists of the data probed both above rivers and above land.

## 3 Results

**Bulk DSD properties for polluted and background clouds**

Given that the aerosol population directly affects cloud formation during the CCN activation process, bulk DSD properties under polluted and background conditions may differ. Figure 6 shows the frequency distribution of the droplet number concentrations (DNC), liquid water content (LWC), and effective diameter ($D_e$) for all measurements inside the plume and under background conditions, irrespective of altitude. Those bulk properties were obtained from the FCDP-measured DSDs. The background clouds presented droplet number concentrations below 200 $cm^{-3}$ for most cases, while being more dispersed for the polluted DSDs. It shows that it is much more likely to find higher DNC under polluted conditions than on background air. This observation may be tentatively justified as an increase in the water vapor competition, which leads to the formation of a higher number of droplets with smaller diameters. However, the water vapor competition is usually discussed for a fixed

LWC, which is not the case for the statistics shown here. The background clouds measured presented lower water contents overall, which could also partly justify the lower concentrations observed.

The effective diameter histograms show distinct droplet sizes distributions for both populations. While around 50% of droplets in the polluted clouds have $D_e$ between 8 and 12 µm, the frequency distribution for the background clouds shows more frequent occurrence of $D_e > 12$ µm, even though they peak at similar diameters. This factor shows that, despite condensing lesser amounts of total liquid water, the background clouds are able to produce bigger droplets than their polluted counterparts. Overall, Figure 6 shows a picture consistent with the water vapor competition concept. However, the DSD formation under a water vapor competition scenario depends on two factors. One is commonly cited on the literature (e.g. Albrecht, 1989) and is related to the impacts on effective droplet sizes as function of aerosol number concentrations. The other factor is how much bulk water the systems are able to condense while the vapor competition is ongoing. Figure 6 suggests that the Manaus pollution plume affects both mechanisms, which are more complex than the water vapor competition process.

An interesting question to address is why LWC is lower for background clouds, i.e., why this type of cloud is relatively inefficient to convert water vapor to liquid droplets. One possible answer is related to total particle surface area in a given volume. Considering a constant aerosol size distribution, when their total number concentration is increased, the total particle surface area per unit volume also increases. In this way there is a wider area for the condensation to occur, leading to higher liquid water contents. Additionally, if there is higher competition for the water vapor, the more numerous and smaller droplets formed under polluted conditions will grow faster by condensation than their background counterparts (because the condensation rate is inversely proportional to droplet size) and will readily reach the threshold for detection by the FCDP (around 1 µm). One point to remember is the high amount of water vapor available during the wet season. Those differences in the bulk condensational growth under polluted or background conditions may explain in part the differences observed in Figures 6c-d, even if the aerosol size distribution changes from the background to the polluted sections. If the bulk condensation is more effective in a polluted environment, it should also lead to increased latent heat release and stronger updrafts. In a stronger updraft the supersaturations tend to be higher, which feeds back into an even higher condensation rate. Other possible physical explanations for the higher LWC in polluted clouds include processes associated with precipitation-sized droplets (i.e., outside the FCDP size range) and aerosol characteristics. If the aerosol-rich plume is able to reduce the effective sizes of the liquid droplets, it will also be able to delay the drizzle formation. In this way, the liquid water would remain inside the cloud instead of precipitating. On the other hand, the fast-growing droplets in the background clouds may grow past the FCDP upper threshold, effectively removing water from the instrument size range. However, the clouds penetrated were predominantly non-precipitating cumulus at early stages of their life cycle. Therefore, the warm-phase was not completely developed and the condensational growth plays a major role in determining the overall DSD properties. The second process identified (i.e. suppressed precipitation staying longer inside the clouds) probably has a lesser impact. The averaged ratio between second moment of the polluted and background DSDs is around 2, which shows that the former have around twice of the total area for condensation than their background counterparts. In this way, the increase in the bulk

condensation efficiency is probably significant. Further studies are encouraged in order to detail and quantify the processes that lead to the observed LWC amount. However, based on Koren et al. (2014), the most determinant factor contributing for the high amount of cloud water under polluted conditions seems to be related to the condensation process. In the referred paper, it is shown that the amount of total condensed water tends to grow with aerosol concentration in a pristine atmosphere.

In order to detail the pollution effects on the total condensation rate and on the DSD properties, averaged properties for different water content and updraft speeds are analyzed. Firstly, given that the LWC is a measure of the total amount of water condensed onto the aerosol population, its correlation with the updrafts should be assessed. The updraft speed at cloud base can be understood as a proxy for the thermodynamic conditions, as it is a result of the meteorological properties profiles in lower levels. In this way, it is possible to disentangle the aerosol and thermodynamic effects by averaging the LWC data at different updraft speed levels. Figure 7a shows the result of this calculation for only the lower 1000 m of the clouds, while also differentiating between polluted and background clouds. The 1000m limit is chosen for both maximizing statistics and also capturing the layer in which the aerosol activation takes place. That layer is possibly thicker under polluted conditions, given the higher availability of nuclei. For similar updraft conditions, i.e., similar thermodynamics, the averaged total liquid water is always higher for polluted clouds. By eliminating the dependence on the thermodynamic conditions, it is possible to conclude that the LWC values are significantly influenced by the aerosol population. This figure shows that, on average, not only are the polluted clouds more efficient at the bulk water condensation but also the resulting LWC scales with updraft speed (linear coefficients, considering the error bars, are 0.13 g s m$^{-2}$ for plume measurements and 0.033 g s m$^{-2}$ for background clouds). In a background atmosphere, most of the aerosols have been activated, and increasing updraft strength does not result in further condensation. On the other hand, the higher availability of aerosols inside the plume allows for more condensational growth as long as enough supersaturation is generated, especially considering that the critical dry diameter for activation is inversely proportional to supersaturation and, consequently, to the updraft speed. However, a deeper analysis in a bigger dataset would be required to assess the statistical significance. The enhanced condensation efficiency and the possible LWC scaling with updraft strength at least partly explain the higher liquid water contents in the plume-affected clouds. The standard deviation bars in Figure 7a indicate that while there is high variability for the LWC in polluted clouds, the clean ones are rather consistent regarding the condensation efficiency.

The water vapor competition effect can be observed by examining droplet effective diameter and number concentrations at a certain LWC interval, as shown in Figures 7b and 7c. In this way, the polluted and background DSD properties can be evaluated irrespective of the bulk efficiency of the cloud to convert water vapor into liquid water. It is clear that, even with the dispersion observed, the two DSD populations present consistently different average behaviors for all LWC intervals. For similar LWC, the averaged effective diameter is always larger on background clouds, with lower droplet number concentrations on average. Those results show a picture clearly consistent with enhanced water vapor competition in polluted clouds. It shows that, given a bulk water content value, droplet growth is more efficient in background clouds. This

process should make background clouds more efficient to produce rain from the warm-phase mechanisms because of the early initiation of the collision-coalescence growth.

Another noteworthy point shown in Figure 7 is the difference between the relationships of $D_e$ and LWC, and of DNC and LWC. While the average effective diameter varies linearly with LWC ($R^2$=0.95 for plume and $R^2$=0.92 for background DSDs), there seems to be a capping on DNC. This means that for low LWC ($< 0.4$ g m$^{-3}$), increases in the total water content are reflected in increased droplet concentrations. For higher LWC values, the averaged DNC remains relatively constant while the effective diameter grows with the water content. This suggests that at low water content levels, i.e., at the early stages of cloud formation, the formation of new droplets has a relatively higher impact on the overall LWC. As the cloud develops, the LWC is tied to the effective diameter of the droplets, as the impact of new droplet formation is weaker at this point. This effect is clearer in background clouds given the limited aerosol availability.

**Vertical DSD development and the role of the vertical wind speed**

The analysis of bulk DSD properties indicates a clear difference between the polluted and background cloud microphysics. However, it is desirable now to further detail those differences. As most of the aerosol activation takes place close to cloud base (Hoffmann et al., 2015), the direct effects of enhancements in particle concentrations should be limited to this region. However, the aerosol effect can carry over to later stages of the cloud life cycle given that it will develop under perturbed initial conditions. One proxy for the cloud DSD evolution in time is to analyze its vertical distribution. For a statistical comparison, a relative altitude for all flights is defined. This relative altitude is calculated as follows: firstly, the closest radiosonde is used in order to obtain the cloud base altitude (as the lifting condensation level) and the freezing level. In case the airplane reached high enough altitudes, its data is instead used to obtain the altitude of the 0ºC isotherm. From those two levels, the relative altitude is calculated as percentages where 0% represents the cloud base and 100% is the freezing level. The altitudes of the cloud base and freezing levels range, respectively, from 100 m to 1200 m and from 4670 m to 5300 m approximately. Three layers are then defined: 1) bottom layer in which relative altitudes vary between 0% and 20%; 2) mid layer for 20% to 50%; and 3) top layer, where the altitude is above 50%. Those specific relative altitude intervals were chosen in order to capture the physics of the cloud vertical structure and to minimize the differences in sample sizes for each layer, as there are more measurements for lower levels. Despite probing individual clouds, the DSD measurements can be combined into the three layers defined and interpreted as representative of a single system. It is conceptually similar to satellite retrievals of vertical profiles of droplets effective radii (e.g. Rosenfeld and Lensky, 1998), where the cloud top radius is measured for different clouds with distinct depths and combined into one profile. This approach was validated with in-situ measurements for the Amazon region by Freud et al (2008).

Figure 8 shows statistical results for the DSDs in the three warm layers defined, while Table 2 shows the respective mean bulk properties. The altitude-averaged values show that the polluted clouds present higher number concentrations and water contents and lower diameters for all layers. Additionally, DNC decays much slower with altitude and droplet growth is significantly suppressed. Those observations point to enhanced collisional growth in the background clouds.

The overall picture of cloud DSD vertical evolution can be seen in Figure 8a. The most discerning feature between the DSDs at different altitudes is related to the concentrations of droplets greater than 25 μm. The concentrations in this size range grow with altitude on average. On the other hand, the concentrations of droplets smaller than 15 μm tend to diminish from the bottom to the top layer. Considering that the vertical dispersion of the DSDs represents at least in part its temporal evolution, this feature is associated with droplet growth where the bigger droplets grow in detriment of the smaller ones. This growth mechanism is the collision-coalescence process, where the bigger droplets collect the smaller ones and acquire their mass. The shaded areas on the figure show that this is not only an average feature, but is also visible in the quantiles.

The statistical results of the vertical evolution of the DSDs are discriminated for the measurements inside the plume and in background regions in Figures 8b-c. At first glance, it is quite clear that the two DSD populations present different behaviors with altitude, meaning that the droplets grow differently depending on the aerosol loading. The plume DSDs present a high concentration on the bottom layer and shows weak growth with altitude. The concentration of small droplets ($< 15$ μm) does not change much with altitude and the top layer DSD is relatively similar to the middle one. On the other hand, the DSDs in the background clouds show a stronger growth with altitude (Figure 8c). The bottom layer DSD presents lower concentrations of small droplets but higher concentrations of bigger droplets than its polluted counterpart does. This coexistence of relatively big and small droplets readily activates the collision-coalescence process, accelerating droplet growth. Comparing both polluted and background DSDs with the overall averages (Figure 8a), it is clear that enhanced aerosol loading leads to less-than-average growth rates and the opposite is true for background clouds. The average growth rate for $D_e$ is 2.90 μm km$^{-1}$ and 5.59 μm km$^{-1}$ for polluted and background clouds, respectively.

The vertical speed inside the cloud is a critical factor as it helps determine the supersaturation and, consequently, the condensation rates in the updrafts. The interaction between the updraft speeds and aerosol loadings ultimately determines the initial DSD formations at cloud base. As mentioned before, the characteristics of the initial DSD may have impacts on the whole cloud life cycle, making the study of the vertical velocities critical for understanding the system development. Figure 9 shows averaged DSDs for different cloud layers and vertical velocities conditions, discriminating between the plume and background cases. The first row shows results for the bottom layer under a) plume and b) background conditions. The mid and top layer results are shown together in the second row, for c) plume and d) background conditions. "Strong" and "Mod" are references to the up- or downdraft speed (strong or moderate). The mid and top layers are considered in conjunction in order to increase the sample size.

For the bottom layer, the vertical velocity has an impact mainly on the concentration of small droplets on polluted DSDs in the range $D < 5$ μm. The regions that presented updrafts are associated with higher concentrations of such droplets because of new droplets nucleated under supersaturation. The downdraft regions mainly contain droplets that already suffered some processing in the cloud system and have relatively lower concentrations of small droplets that were probably collected by bigger ones. Additionally, small droplets ascend readily with the updrafts given their low mass, which is also a factor that can contribute to the differences between up- and downdraft DSDs. However, the dispersion shown in the shaded areas shows that the populations of DSDs in up- and downdrafts are relatively similar, suggesting a homogeneous layer with

respect to DSD types. The DSDs shown on Figure 9a indicate single-mode distributions, which hampers collection processes and explains the similarities between the different vertical velocity regions. On the other hand, the background clouds have a second mode, especially in the downdrafts given the additional cloud processing, which favor the collision-coalescence process. The particles associated with background air in the Amazon are not only less numerous but also bigger overall compared to the urban pollution, and both of those features favor faster growth by condensation because of less vapor competition and larger initial sizes. It is interesting to note that the background DSDs in the strong updraft regions are narrower when compared to their polluted counterpart. In a polluted environment, there is not only the natural background aerosol population but also the urban particles emitted from Manaus. The mixture of the two, with the consequent physicochemical interactions, permits the formation of droplets over a wider size range, with a prolonged tail towards the lower diameters. The shaded areas show that the differences between the DSDs in the up- and downdraft regions are statistically relevant for the background clouds and are not a mere averaging feature.

Cloud droplets keep growing as they move to higher altitudes, but the way in which it occurs is rather different in a background or plume-affected environment. For polluted DSDs, there are two modes at the higher altitudes: one reminiscent of the lower levels and the other is probably mainly a result of additional condensational growth. In those systems, the additional processing does not seem to be effective to produce bigger droplets, as shown by the blue line and shaded area in Figure 9c. For the background clouds, DSDs in the updraft regions show similar modes to their polluted counterparts, one close to 10 μm and the other at around 18 μm. However, there are appearances of droplets bigger than 30 μm that contribute to the formation of a third mode in the mid and top layers. This mode appears on the strong downdraft regions, which suggests it appears after in-cloud processing.

**4 Summary and conclusions**

This study focused on the analysis of microphysics of warm-phase clouds in Amazonia during the wet season, with a specific emphasis on interactions with the pollution emitted by Manaus city. A statistical approach was used to compare several clouds probed in different flights on different days. Concerning the effects of the pollution plume on the cloud DSDs bulk properties, there are two processes to consider. A polluted environment with high particle count presents a high total area for the condensation, favoring higher bulk liquid water on the DSDs. Additionally, the total amount of condensed water scales with updraft speed in the plume-affected clouds, which is not the case for background clouds. The growth processes under background aerosol levels are much more effective even with lower bulk liquid water contents. Despite the lower amount of water condensed in background DSDs, bigger droplets readily form given the early start of the collision-coalescence process (which does not increase LWC). Polluted clouds had droplets 10%-40% smaller on average and more numerous droplets (as high as 1000% difference) in the same vertical layers inside the cloud.

The averaged DSDs in different layers of warm clouds show droplets grow with altitude overall, with bigger droplets acquiring mass from the smaller ones. However, the growth rates with altitude are much slower for plume-affected clouds

(almost half of the clean growth rate) due to the enhanced water vapor competition and the lack of bigger droplets at the onset of the systems. Background clouds present relatively high concentrations of droplets greater than 20 μm near cloud base that contributed to the growth rates, especially taking into account the non-linear nature of the collection process. With respect to warm-phase cloud DSDs, the updraft strength does not seem to be the major driving force for effective droplet growth, especially beyond the 20 μm range. The most important features to produce such big droplets are weak water vapor competition (usually observed in background clouds) and the existence of bi-modality at the lower levels of the cloud. The weak water vapor competition favors the formation of big droplets (> 20 μm) required for the collision-coalescence process, while the bi-modality favors the efficiency of the collision-coalescence process due to the large terminal velocity differences between the modes. However, the thermodynamic role of the updraft speeds should not be underestimated. It is responsible for transporting hydrometeors beyond the freezing level, activating the cold processes. Those processes are known to be associated to thunderstorms and intense precipitation. Nevertheless, the main feature that determines warm-phase DSD shapes seems to be the aerosol conditions, with the vertical velocities playing a role in the modulation of the distributions. While the effects of aerosol particles in the warm layer of the clouds are relatively straightforward, this may not be the case for the mixed and frozen portions. An aspect that was not directly addressed in this work is the impacts that warm layer characteristics have on the formation of the mixed phase (above the 0°C isotherm). Given that aerosols alter the properties of the whole warm phase, it is reasonable to assume that this would have an impact on the initial formation of the mixed layer. Such impacts can be in the form of the timing and physical characteristics of the first ice particles and the maximum altitude with supercooled droplets above the freezing level. This issue will be addressed in future studies, taking advantage of data provided by the HALO (High Altitude and Long Range Aircraft) airplane that operated in the second GoAmazon2014/5 IOP between September and October, 2014.

*Acknowledgements:* This work was funded by FAPESP (project Grant 2014/08615-7 and 2009/15235-8), and the Atmospheric Radiation Measurement (ARM) Climate Research Facility, a U.S. Department of Energy Office of Science user facility sponsored by the Office of Biological and Environmental Research. We acknowledge the support from the Central Office of the Large Scale Biosphere Atmosphere Experiment in Amazonia (LBA), the Instituto Nacional de Pesquisas da Amazonia (INPA), and the Instituto Nacional de Pesquisas Espaciais (INPE). The work was conducted under 001262/2012-2 of the Brazilian National Council for Scientific and Technological Development (CNPq). J. Fan was supported by the U.S. Department of Energy Office of Science Atmospheric System Research (ASR) Program. We thank Karla M. Longo for her leadership in the Brazilian side of the GoAmazon2014/5.

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

## Figure captions

**Figure 1**: Conceptual schematic for the flight patterns planning. It shows Manaus city and its pollution plume dispersing over the surrounding Amazon forest. The Cu field shown is very common during the wet season and is representative for most of the cloud conditions during the flights. The yellow circles indicate a 100 km radius from Manaus airport, although

the figure is not meant to be quantitatively accurate. The lines with arrow heads show the most common flight plan used, where blue regions are possible locations for the background air measurements and the red ones indicate measurements inside the plume section (dashed white lines). T3 is a GoAmazon site to the north of Manacapuru.

**Figure 2:** Trajectories for all G-1 flights during GoAmazon2014/5 IOP1. Manaus is located close to the {-60, -3} point, marked with an "X", while the T3 site is marked with the black circle.

**Figure 3**: CN concentrations around Manaus for 10 March 2014. θ is the azimuth angle and is zero for East direction and grows counterclockwise. Colors are proportional to the horizontal distance (km) between Manaus airport and the aircraft. The black dots represent the angular mean CN concentration for each one of the 60 bins (azimuth). The vertical dashed lines represents the limits of the plume location.

**Figure 4**: The same as Figure 2, with the coloring representing the plume classification for 10 March 2014. The green-

colored dots represent unclassified points, red is for plume, and cyan is for background conditions. The inset shows the median (cyan) and the 25% (Blue) and 90% (red) percentiles profiles of CN concentrations.

**Figure 5:** CCN concentrations as function of supersaturation. Measurements inside the plume are shown in red, while background conditions are represented in blue.

**Figure 6:** Normalized histograms of cloud droplets properties affected and unaffected by the Manaus plume. (a-b) Total

droplet number concentrations ($cm^{-3}$), (c-d) liquid water content ($gm^{-3}$), and (e-f) effective diameter (µm).

**Figure 7**: Mean (a) LWC values for different log-spaced w intervals and mean $D_e$ (b) and DNC (c) for log-spaced LWC intervals. Error bars are the standard deviation for each interval. Blue points indicate background measurements, while red

ones are relative to the polluted ones. The points are located at the middle of the respective bin intervals. Those results are limited to the first 1000 m of the clouds.

**Figure 8**: Averaged DSDs for three different cloud layers - bottom, mid and top of the warm layer. Graph (a) shows the results for all DSDs irrespective of classification, while (b) is for polluted DSDs only, and (c) for background. Lines represent averages, while the shaded areas represent the dispersion between the 25% and 75% quantiles.

**Figure 9**: Averaged DSDs as function of altitude, presence of up/downdrafts, and aerosol conditions. The first row shows results for the bottom layer under (a) polluted and (b) background conditions. The mid and top layers results are shown together in the second row for (c) plume and (d) background conditions. "Strong Down" means the presence of strong downdrafts, with velocities lower than -2 m s$^{-1}$. "Mod Down" is moderate downdrafts, with -2 m s$^{-1}$ < $w$ ≤ 0. "Mod Up" and "Strong Up" are the equivalents for updrafts. Their velocities ranges are, respectively, 0 < w ≤ 2 m s$^{-1}$ and w > 2 m s$^{-1}$. The shaded areas represent the dispersion between the 25% and 75% for the strong downdrafts (in blue) and updrafts (in red).

**Table captions**

**Table 1:** Dates and times for all G-1 flights during GoAmazon2014/5 IOP1. Local time for Manaus is UTC-4. All flights were carried out in the year 2014.

**Table 2:** Averaged bulk DSD properties for the three warm-phase layers and the respective standard deviations. The bottom layer is defined by relative altitudes between 0% and 20%, the mid layer between 20% and 50% and the top between 50% and 100%.

**Tables**

**Table 1:** Dates and times for all G-1 flights during GoAmazon2014/5 IOP1. Local time for Manaus is UTC-4. All flights were carried out in the year 2014.

| Flight Number | Date | Start Time (UTC) | End Time (UTC) |
|:---:|:---:|:---:|:---:|
| 1 | February 22 | 14:38:27 | 17:25:26 |
| 2 | February 25 | 16:32:06 | 18:40:07 |
| 3 | March 1 | 13:35:37 | 15:27:35 |
| 4 | March 1 | 17:18:48 | 18:47:07 |
| 5 | March 3 | 17:46:34 | 19:11:57 |
| 6 | March 7 | 13:09:51 | 15:35:25 |
| 7 | March 10 | 14:26:37 | 17:09:35 |
| 8 | March 11 | 14:42:23 | 17:51:08 |
| 9 | March 12 | 17:21:25 | 19:29:42 |
| 10 | March 13 | 14:16:09 | 17:21:27 |
| 11 | March 14 | 14:18:54 | 16:48:23 |
| 12 | March 16 | 14:40:17 | 17:26:32 |
| 13 | March 17 | 16:24:40 | 19:26:36 |
| 14 | March 19 | 14:26:38 | 17:17:48 |
| 15 | March 21 | 16:33:47 | 18:56:07 |
| 16 | March 23 | 14:59:05 | 17:43:34 |

**Table 2:** Averaged bulk DSD properties for the three warm-phase layers and the respective standard deviations. The bottom layer is defined by relative altitudes between 0% and 20%, the mid layer between 20% and 50% and the top between 50% and 100%.

| Layer | DNC (cm$^{-3}$) | | $D_e$ (µm) | | LWC (g m$^{-3}$) | |
|:---:|:---:|:---:|:---:|:---:|:---:|:---:|
| | Plume | Background | Plume | Background | Plume | Background |
| **Bottom** | $317 \pm 190$ | $127 \pm 131$ | $11.3 \pm 2.00$ | $14.2 \pm 4.19$ | $0.206 \pm 0.216$ | $0.114 \pm 0.122$ |
| **Mid** | $360 \pm 276$ | $81.6 \pm 77.4$ | $17.7 \pm 4.12$ | $18.4 \pm 6.18$ | $0.848 \pm 0.788$ | $0.183 \pm 0.218$ |
| **Top** | $191 \pm 203$ | $7.64 \pm 14.9$ | $15.5 \pm 5.28$ | $31.7 \pm 4.12$ | $0.522 \pm 0.703$ | $0.0766 \pm 0.151$ |

**Figures**

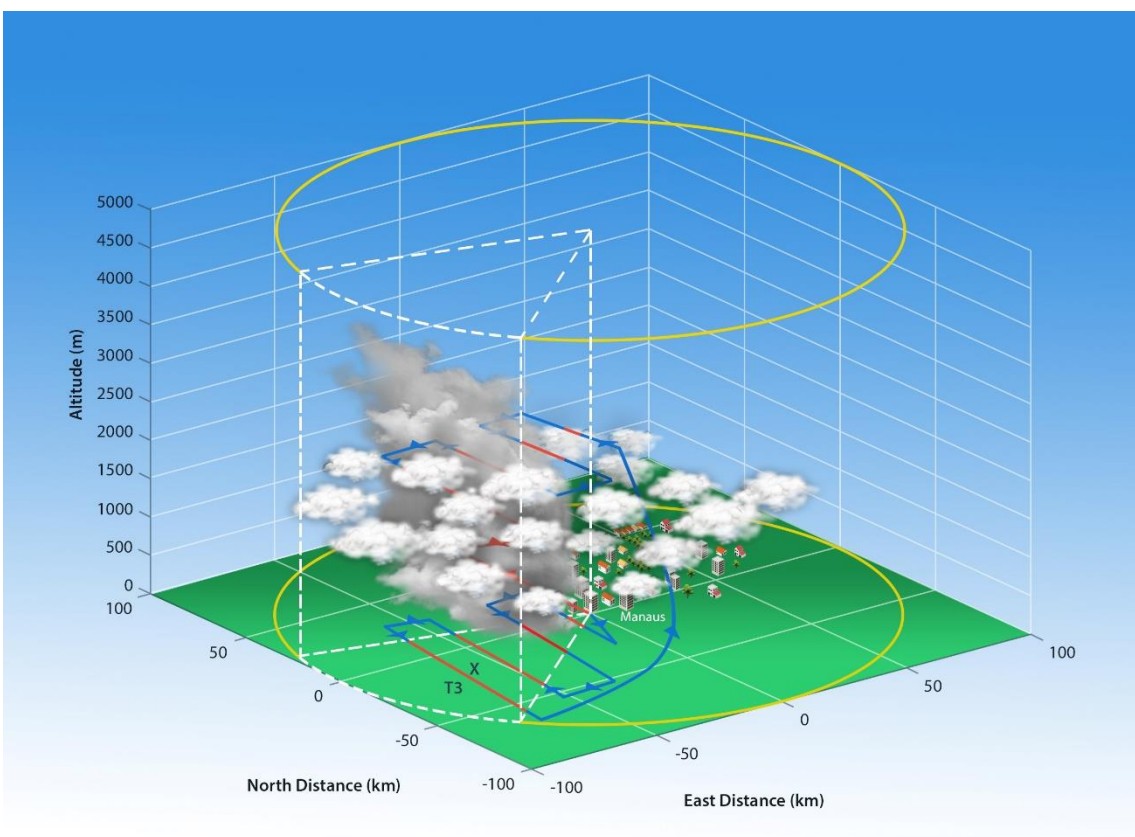

**Figure 1:** Conceptual schematic for the flight patterns planning. It shows Manaus city and its pollution plume dispersing over the surrounding Amazon forest. The Cu field shown is very common during the wet season and is representative for most of the cloud conditions during the flights. The yellow circles indicate a 100 km radius from Manaus airport, although the figure is not meant to be quantitatively accurate. The lines with arrow heads show the most common flight plan used, where blue regions are possible locations for the background air measurements and the red ones indicate measurements inside the plume section (dashed white lines). T3 is a GoAmazon site to the north of Manacapuru.

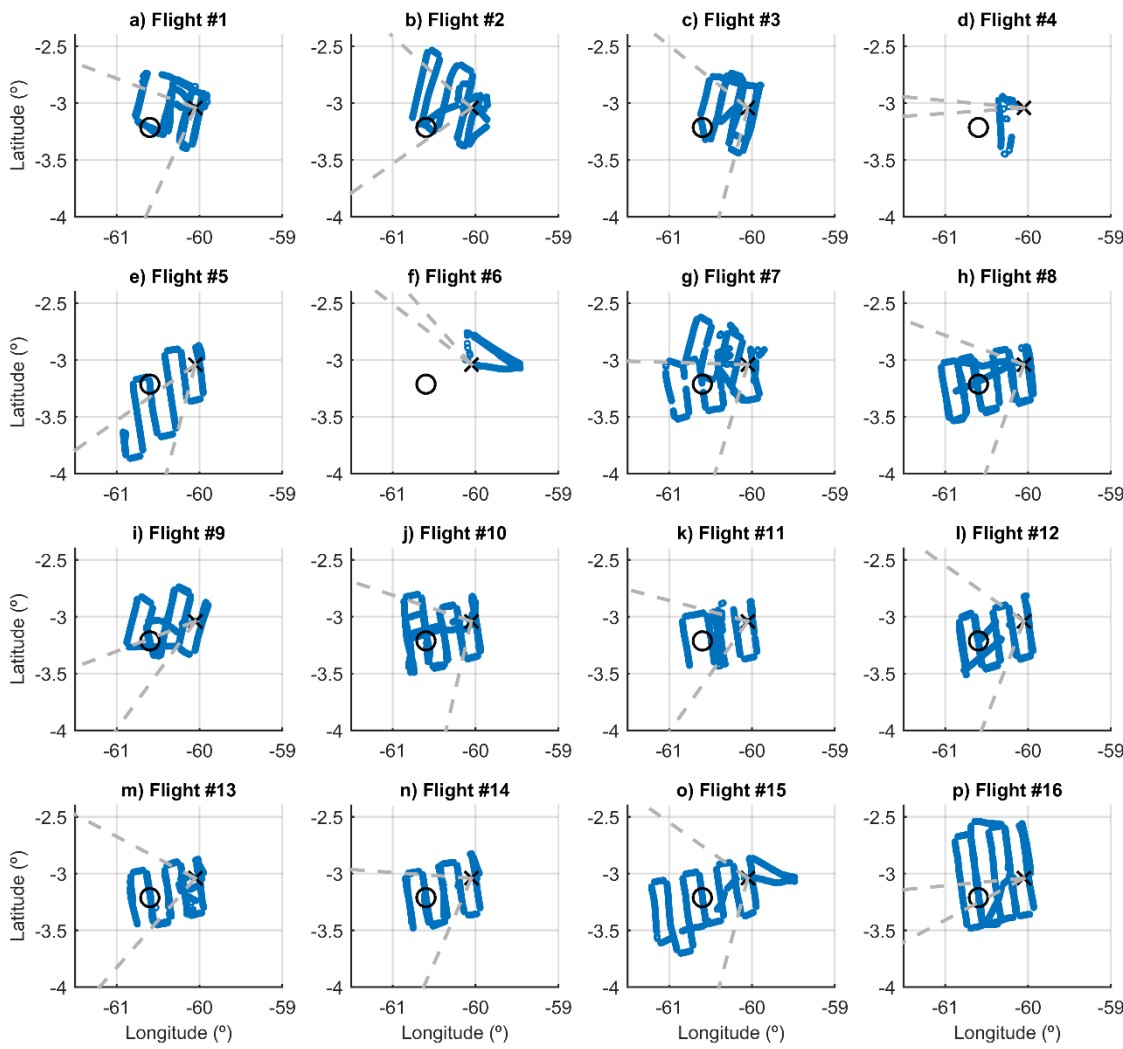

**Figure 2:** Trajectories for all G-1 flights during GoAmazon2014/5 IOP1. Manaus is located close to the {-60, -3} point, marked with an "X", while the T3 site is marked with the black circle.

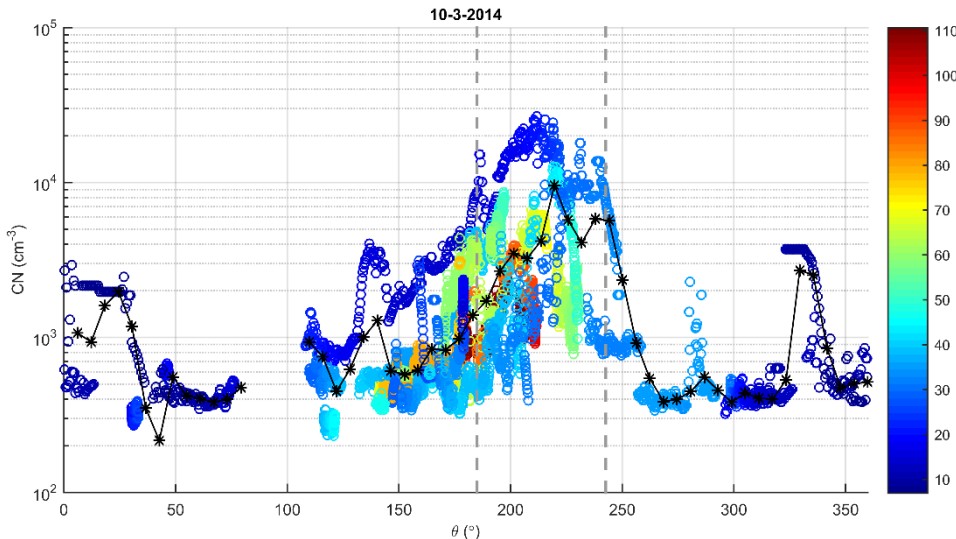

**Figure 3**: CN concentrations around Manaus for 10 March 2014. θ is the azimuth angle and is zero for East direction and grows counterclockwise. Colors are proportional to the horizontal distance (km) between Manaus airport and the aircraft. The black dots represent the angular mean CN concentration for each one of the 60 bins (azimuth). The vertical dashed lines represents the limits of the plume location.

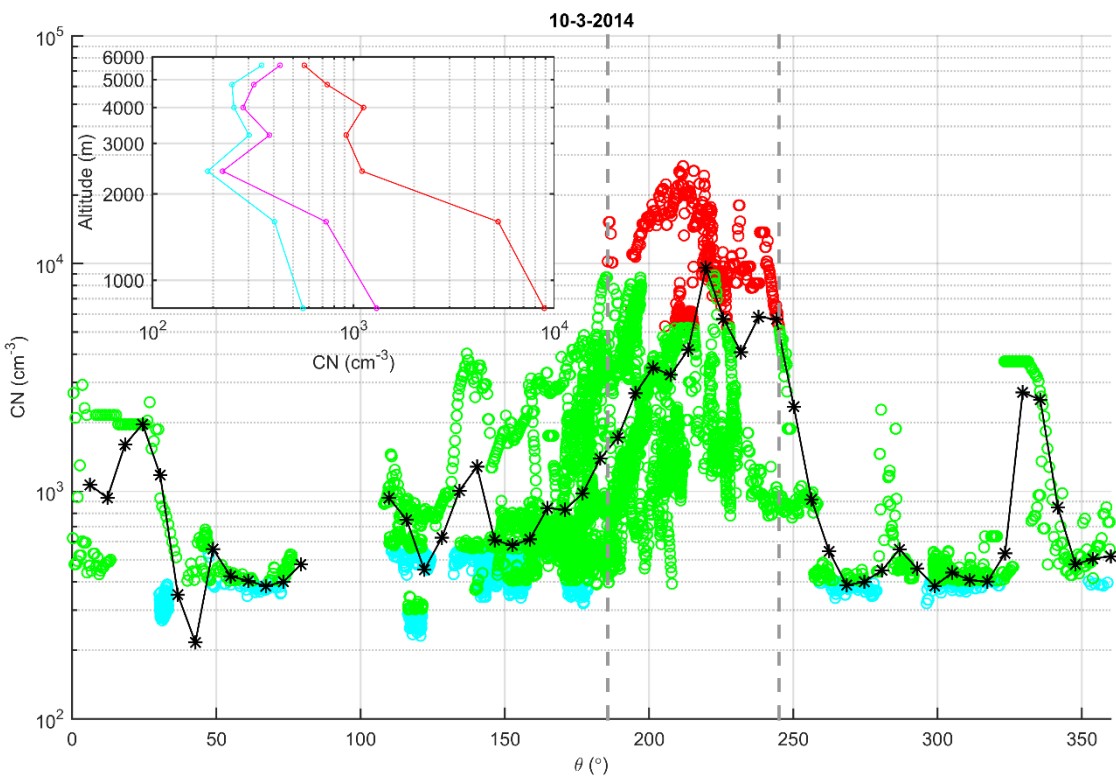

**Figure 4**: The same as Figure 2, with the coloring representing the plume classification. The green-colored dots represent unclassified points, red is for plume, and cyan is for background conditions. The inset shows the median (cyan) and the 25% (blue) and 90% (red) percentiles profiles of CN concentrations.

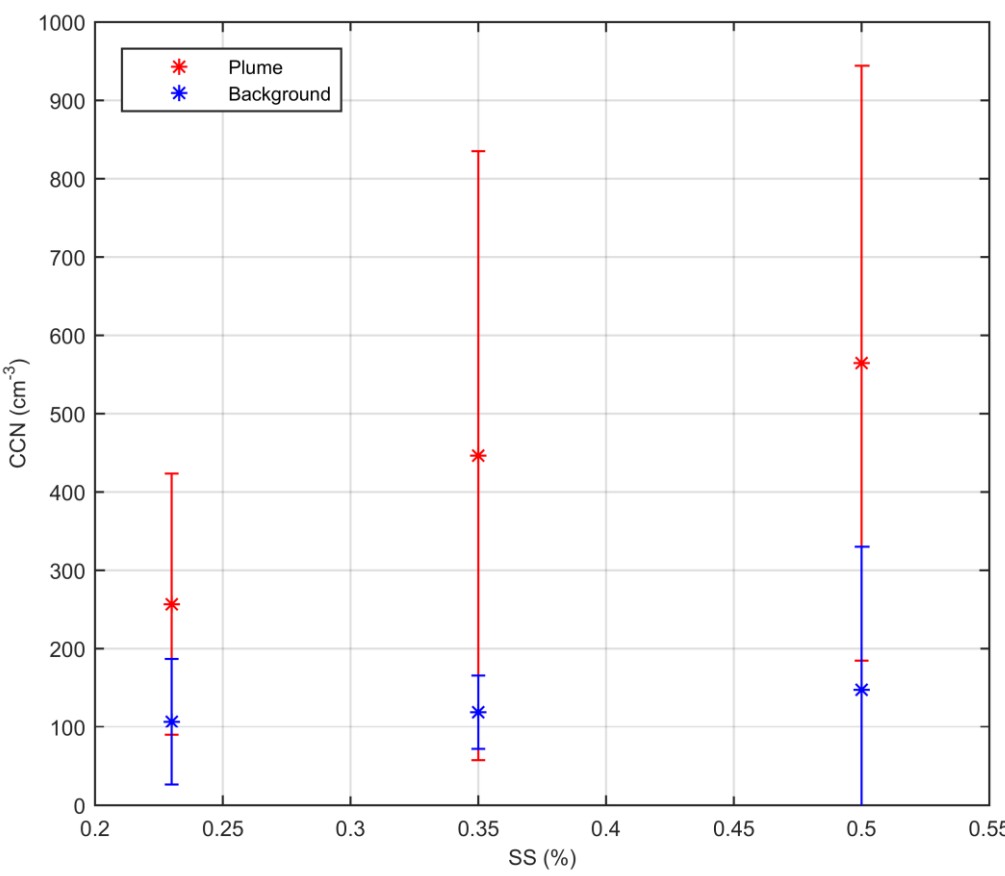

**Figure 5:** CCN concentrations as function of supersaturation. Measurements inside the plume are shown in red, while background conditions are represented in blue.

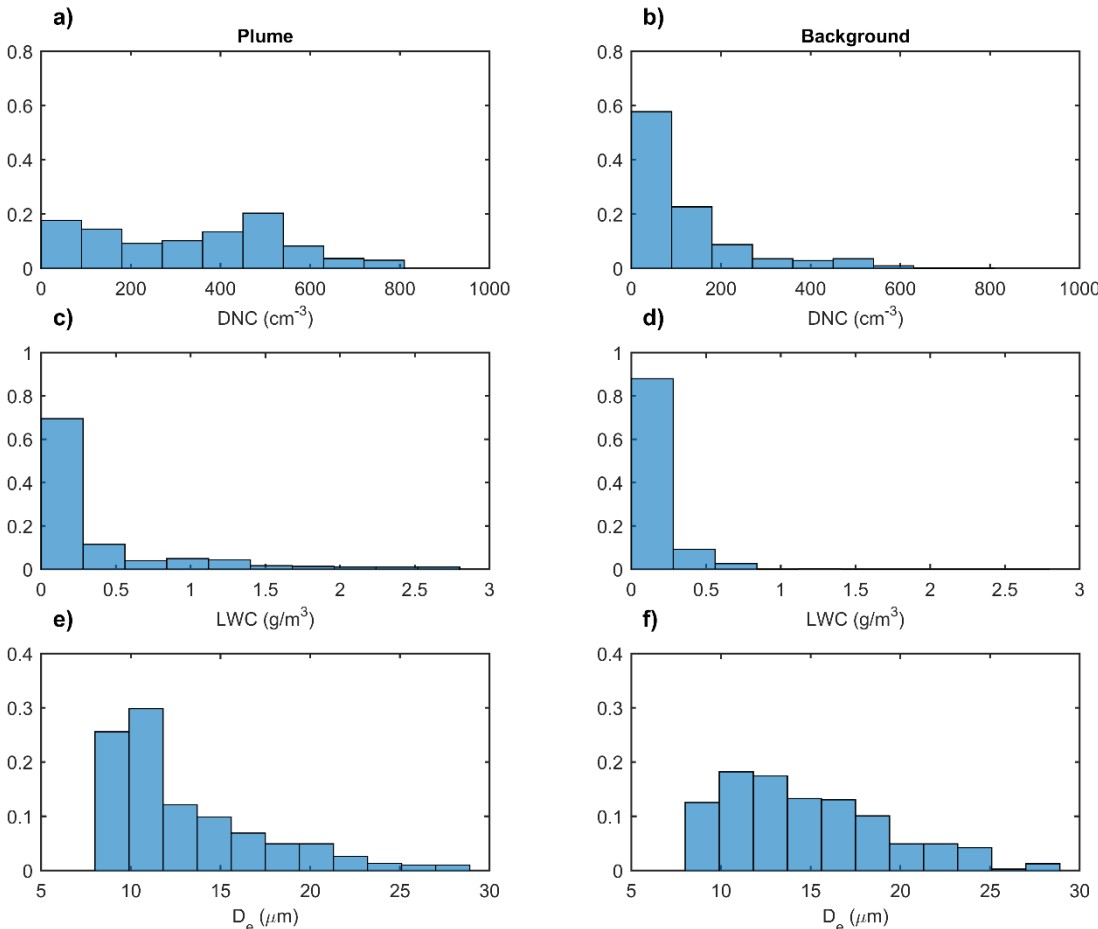

**Figure 6:** Normalized histograms of cloud droplets properties affected and unaffected by the Manaus plume. (a-b) Total droplet number concentrations (cm$^{-3}$), (c-d) liquid water content (gm$^{-3}$), and (e-f) effective diameter (µm).

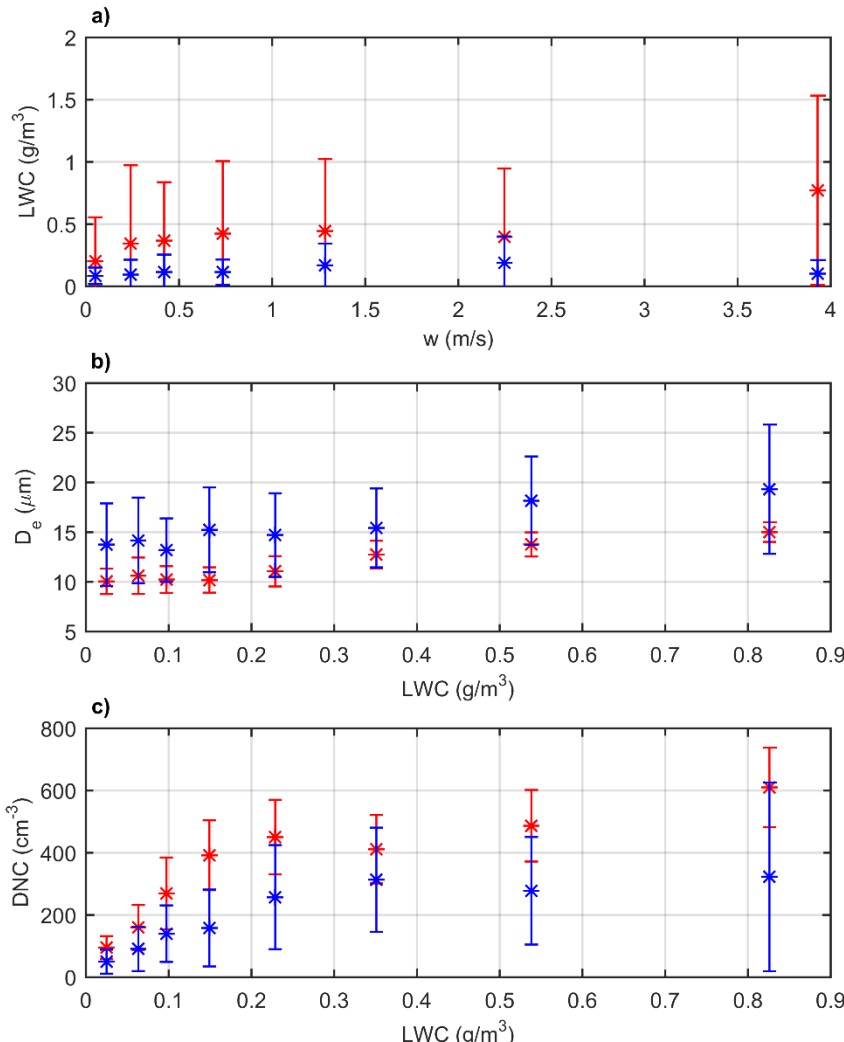

**Figure 7**: Mean (a) LWC values for different log-spaced w intervals and mean $D_e$ (b) and DNC (c) for log-spaced LWC intervals. Error bars are the standard deviation for each interval. Blue points indicate background measurements, while red ones are relative to the polluted ones. The points are located at the middle of the respective bin intervals. Those results are limited to the first 1000 m of the clouds.

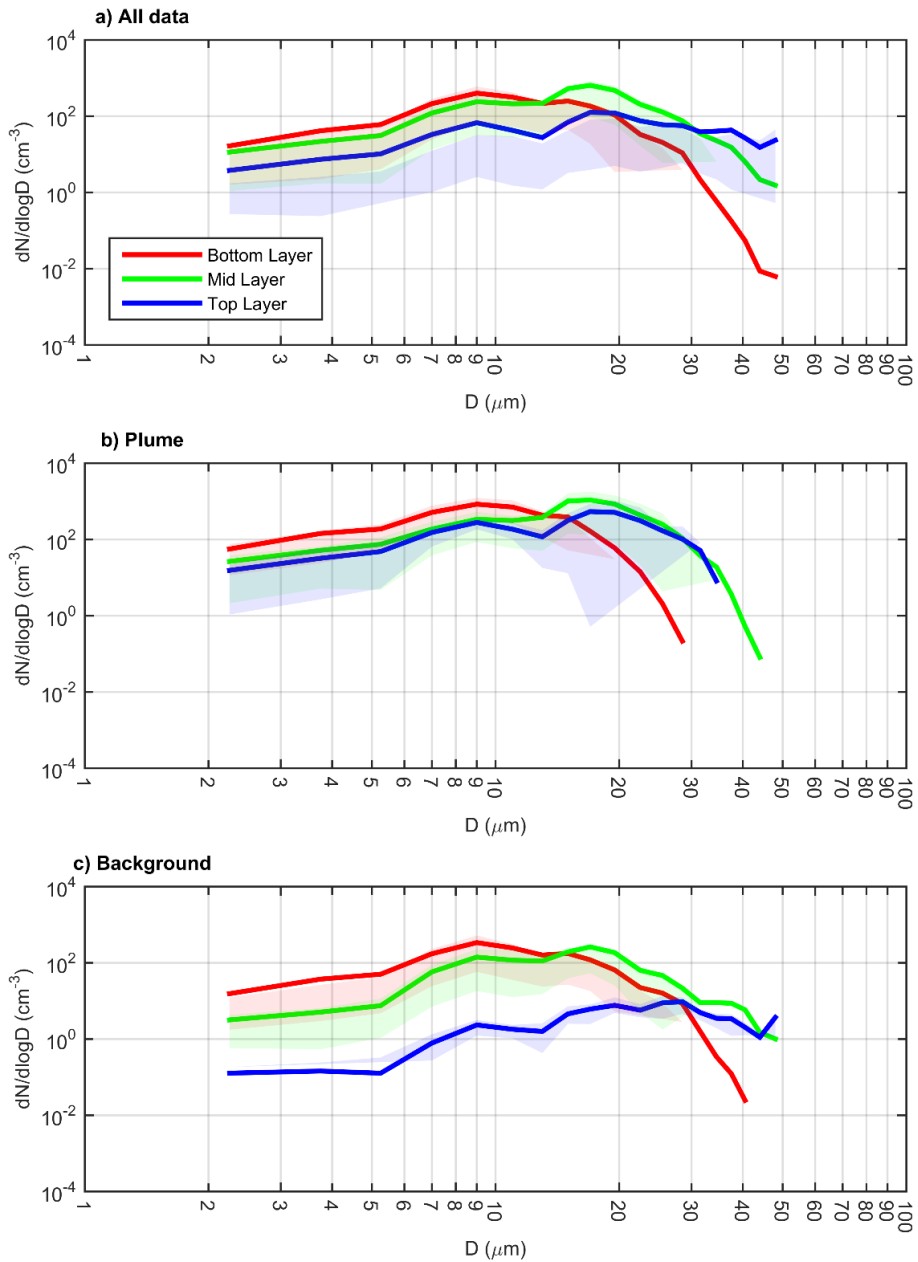

**Figure 8**: Averaged DSDs for three different cloud layers of bottom, mid and top of the warm layer. Graph (a) shows the results for all DSDs irrespective of classification, while (b) is for polluted DSDs only and (c) for background. Lines represent averages, while the shaded areas represent the dispersion between the 25% and 75% quantiles.

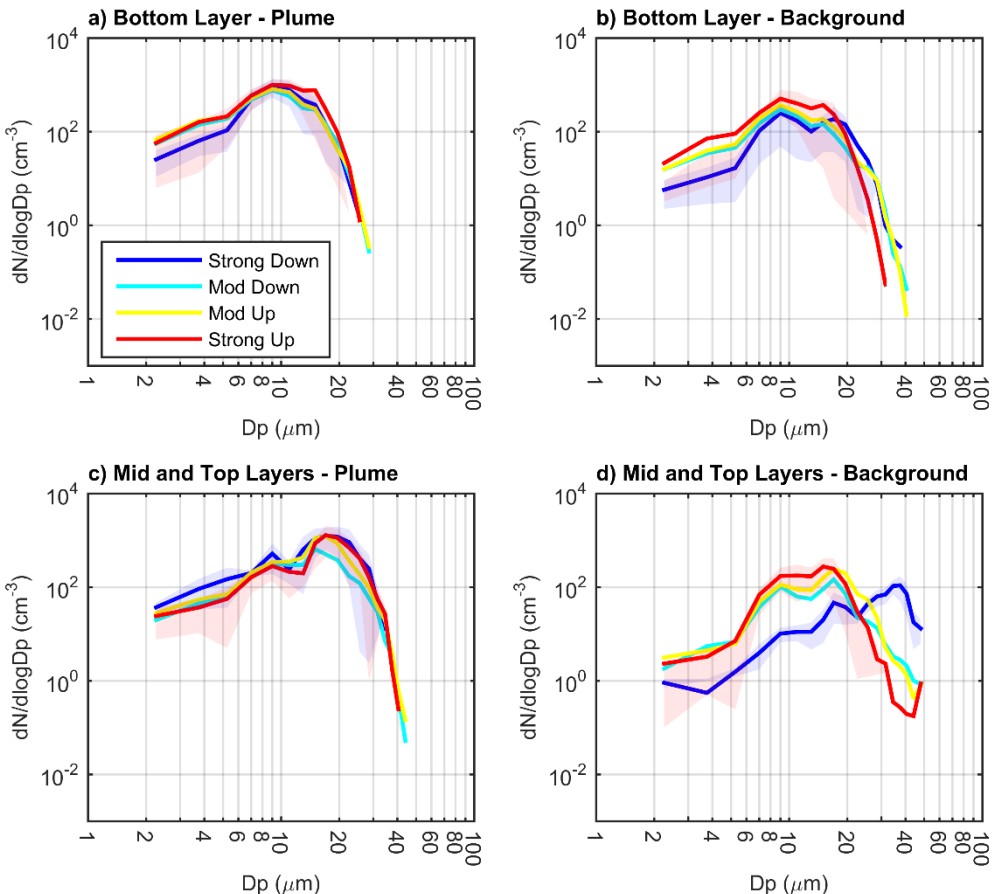

**Figure 9**: Averaged DSDs as function of altitude, presence of up/downdrafts and aerosol conditions. The first row shows results for the bottom layer under (a) polluted and (b) background conditions. The mid and top layers results are shown together in the second row for (c) plume and (d) background conditions. "Strong Down" means the presence of strong downdrafts, with velocities lower than -2 m s$^{-1}$. "Mod Down" is moderate downdrafts, with -2 m s$^{-1}$ < $w$ ≤ 0. "Mod Up" and "Strong Up" are the equivalents for updrafts. Their velocities ranges are, respectively, 0 < $w$ ≤ 2 m s$^{-1}$ and $w$ > 2 m s$^{-1}$. The shaded areas represent the dispersion between the 25% and 75% percentiles for the strong downdrafts (in blue) and updrafts (in red).