# Peer review of "Impacts of the Manaus pollution plume on the microphysical properties of Amazonian warm-phase clouds in the wet season"

_Atmospheric Chemistry and Physics, 2015_

## Referee Comment (RC1) · Anonymous Referee #1 · 9 Mar 2016

The study under review applied in-situ instrumental data from an aircraft campaign with the aim to seperate cloud microphysical properties for conditions of clean/background air and for air polluted from anthropogenic aerosol. Therefore the aircraft measurements were taken inside and outside of a plume created from the city Manaus during the wet season over the Amazon regions. While studies for the dry period exist, the study seems to be one of the first to investigate the wet season, which has the advantage that background conditions are comparable to oceanic conditions, i.e. thermodynamic variables do not vary much with distance from the city.

The study clearly outlines and justifies the methods that are applied. The results are presented in a clear way and the discussion is conclusive and supports the current

understanding of aerosol-cloud interactions. The data is unique and the results should definitely be of interest for the scientific community.

Therefore I would suggest to publish the manuscript after some minor corrections, which are listed in the following.

**Minor revisions:**

1. You could consider to seperate the results section into topic related subsections, as the section is quite long in general (E.g. differences in LWC for background and polluted clouds, effect of updraft speed, vertical development of clouds). This would make it easier for the reader to find the relevant information in the results.

2. The summary and conclusions section ends quite abrupt. Consider to add a short outlook. What are the remaining open questions? Are further field studies planned? You already mention that the effects on ice-clouds is one focus for future endeavors in the motivation.

p.4, l.13: Can you tell more about the uncertainties of the instruments? E.g. what is the accuracy of the particle concentrations from the CPC?

p.6, l.27: You name one factor is commonly cited in literature but do not add any references. I suggest to add some references at this point.

p.7, l.18: Calculations show that . . . → It would be nice if you shortly can present how you did this estimation.

p.8.,l. 21: While your statement seems to be true for the background clouds, especially for the polluted clouds there seems to be an increase in the last LWC bin. Also the spread is increased. Can the latter be explained by a larger LWC bin size?

Table 2: Add the definition of bottom, mid and top layer to the Table caption.

Figure 2: This figure looks a bit clumsy. My suggestion would be to create a plot with subfigures with the individual flight plans and add the estimated plume area and the

average wind direction for each flight.

**Phrasing / spelling corrections:**

p.1, l.11: in terms *of* aerosol conditions

p.1, l17: split the sentence after the brackets → The cloud droplets observed are in the range . . .

p.1, l.24: correct superscript of km-1

p.1, l.25: Why you use *e.g.* for the definition of larger droplets? In my opinion, you can just omit this.

p.1, l.25: to *the* cloud base

p.1., l.26: change sentence structure to: The overall shape of the droplet size distribution (DSD) does not appear to be . . .

p.1, l.31: initiation of *the* collision-coalescence

p.2, l.4: maintain*s*

p.2, l.8: for the Amazon *by* Martin et al.

p.2, l.11: that a city like Manaus *has* on atmospheric conditions

p.2, l.22: Amazonian cloud properties

p.3, l.7: suggests → suggest

p.3, l.8: over *the* Manaus area

p.3, l.9: stronger wind component → dominant wind component

p.3, l.20: clouds microphysical properties → cloud microphysical properties

p.3, l.25: add comma after background air reference

p.4, l.1: consider to change pollution-aerosols to anthropogenic aerosols

p.4, l.2: are almost only urban, while biomass-burning contribution is very exceptional

p.4, l.6: omit numbered before chronologically

p.4, l.29: what is meant by true airspeed? I guess you mean the speed of the aircraft?

p.6, l.22: by effective size you refer to effective diameter $D_e$?

p.7, l.29: omit *profiles*

p.7, l.30: updraft speeds levels → updraft speed levels

p.8, l.18: relationships De x LWC and DNC x LWC → relationships of De and LWC, and of DNC and LWC

p.8, l.20: omit *e.g.*

p. 9, l.4: omit brackets. Instead write: for each layer, as there are more measurements for lower levels.

p.9, l.14: its mass → their mass

p.9, l.19: omit *e.g.*

p.9, l.32: and l.33: once you write plume and once polluted. Try to be consistent.

p.10, l.10: justifies → explains

p.10, l.11: vertical velocities region → vertical velocity region

p.10, l.25: the updraft regions DSD → DSDs in the updraft region

p.11, l.6: Polluted clouds had 10

p.11, l.16: omit *e.g.*

p.11, l.17: bi-modality *favors* the efficiency

p.11, l.20: aerosols conditions → aerosol conditions

Figure 5 caption: affected or not → affected and unaffected

Figure 5 caption: units of LWC should be gm$^{-3}$

Figure 6 caption: add the information that this is for clouds lower than 1000 m only

[Figure]

---

## Referee Comment (RC2) · Anonymous Referee #2 · 10 Mar 2016

The manuscript describes unique aircraft measurements of polluted and pristine clouds over the Amazon region during the wet season of 2014. The results are of potential interest not only for those involved in the GoAmazon experiment but to the general ACP audience.

However, the manuscript is poorly written and needs substantial revision to meet ACP standards. Part of the methodology should be better explained and some of the results needs further investigation. Moreover, I have serious concerns about some methods and the interpretation of results. Hence, I cannot recommend its publication without a major revision.

I have annotated the author's PDF file with many comments and questions to the au-

<Printer-friendly version>

thors, which I hope will help to improve the manuscript. Here, I only list my major concerns.

1) Introduction needs to be throughly revised. Lines discussing this work are mixed with paragraphs discussing the current state of the art, making it hard to follow for those not part of GoAmazon.

2) Session about the instrumentation should explain what corrections or data processing were performed for the different instruments / probes used. Alternatively, other papers describing that should be cited.

3) Authors used CN to identify if clouds probed under each circunstance were or were not beeing affected by the plume of pollution.

I beleive CCN would be better to indicate the influence of the plume on the clouds for 2 reasons. Firstly, because most of the initial pollution particles emitted will be too small to become CCN, hence the initial plume will not affect much the cloud formation. Secondly, as the plume is chemically and physically transformed downwind of Manaus, the extra aerosols will grow, be oxidized, and thus will interfere more and more with the CCN population. See for instance previous results from Kuhn et al (2010). Hence, as the G1 flight legs are at different distances from Manaus, CCN would be a better indicator than CN.

4) For selecting the in-plume events, the authors defined a cone where the plume was most likely to be found.

This cone was centered at the airport as if the pollution plume were being dispersed from that single position. While this approch might work for larger distances, for the short distances from Manaus (closer legs) the airport-angle will not confine the plume.

5) The authors based their whole analyses on a bold hypothesis:

Given the nature of the meteorology in the Amazonian wet season, i.e. (...) horizontal homogeneity, there is no significant difference between the thermodynamic conditions

inside and outside the plume region (...).

This would be true only for Amazon regions with an uniform vegetation cover, which is not the case at all for the region of Manaus. The Manaus plume goes towards T3, which is the direction of the Solimoes river. Hence, the in-plume cases studied are mostly over or close to the river. On the other hand, the out-plume clouds are far away from the plume and hence from the river. Therefore, as we know from previous studies that the river breeze is significant, one cannot assume that the thermodynamics are the same (over the river and far away)!

To assess the validity of their hypothesis, the authors could, for instance:

- use radiosondes close and away from the river

- look at the specific humidity around the clouds (polluted vs pristine)

- verify the average time of day when polluted vs pristine clouds were sampled

- verify the location (lat/lon) where the polluted vs pristine clouds were sampled

- etc...

6) When authors look at DSD from different altitudes, they divide the vertical from LCL (0%) to freezing level (100%). Then they made averages for relative altitude ranges of 0-20, 20-50 and >50%.

There are two things going on. Firstly, the G1 samples are not well distributed in the vertical, hence the authors had to choose uneven limits to get the same number of samples in each. However, not all shallow clouds will develop as high as the freezing level.

Therefore, the average for the bottom layer includes some clouds that did not extend at altitudes >20% and more clouds that did not develop > 50%. On the contrary, samples for the top layer are, by definition, all from clouds that extended from the LCL up to > 50%.

Interactive
comment

[Figure]

Hence, this introduces a large bias. It is mixing clouds of different total vertical development, in different amounts, in each of the three categories. Hence one could not compare DSD from different altitudes, just DSDs from the same altitude for polluted/pristine cases.

Please also note the supplement to this comment:
http://www.atmos-chem-phys-discuss.net/acp-2015-1049/acp-2015-1049-RC2-supplement.pdf

**Supplement:**

[Figure]

[revised manuscript text omitted]
. The distributions peak at different diameters, with the modal $D_e$ being larger in background conditions. This factor shows that, even with lesser amounts of total liquid water, the background clouds are able to produce bigger droplets than their polluted counterparts. Overall, Figure 5 shows a picture consistent with the water vapor competition concept. However, the DSD formation under a water vapor competition scenario depends on two factors. One is commonly cited on the literature and is related to the impacts on effective droplet sizes as function of aerosol number concentrations. The other factor is how much water there is for the aerosol population to compete on. Figure 5 suggests that the Manaus pollution plume affects both mechanisms, and the analysis is more complex than the water vapor competition process.

[revised manuscript text omitted]

---

## Author Response (AR1)

**COMMENTS RECEIVED FROM REVIEWER #1**

**General comments**

The study under review applied in-situ instrumental data from an aircraft campaign with the aim to seperate cloud microphysical properties for conditions of clean/background air and for air polluted from anthropogenic aerosol. Therefore the aircraft measurements were taken inside and outside of a plume created from the city Manaus during the wet season over the Amazon regions. While studies for the dry period exist, the study seems to be one of the first to investigate the wet season, which has the advantage that background conditions are comparable to oceanic conditions, i.e. thermodynamic variables do not vary much with distance from the city. The study clearly outlines and justifies the methods that are applied. The results are presented in a clear way and the discussion is conclusive and supports the current understanding of aerosol-cloud interactions. The data is unique and the results should definitely be of interest for the scientific community. Therefore I would suggest to publish the manuscript after some minor corrections, which are listed in the following.

**Authors answer**

Dear Reviewer #1, we would like to express our gratitude for your efforts to review the submitted manuscript. We found your comments very important to improve the quality of the manuscript. Following you will find a detailed explanation of our approach regarding your questions. Your concerns are numbered and answered individually in order in the next section of this document.

Thank you and best regards,

Micael A. Cecchini and coauthors

**Minor revisions:**

1.
    a. **(Question)** You could consider to separate the results section into topic related subsections, as the section is quite long in general (E.g. differences in LWC for background and polluted clouds, effect of updraft speed, vertical development of clouds). This would make it easier for the reader to find the relevant information in the results.

    b. **(Answer)** We agree that it would be clearer to separate the results into subsections. We divided into two separate subsections, labeled "Bulk DSD

properties for polluted and background clouds" and "Vertical DSD development and the role of the vertical wind speed".

2.

   a. **(Question)**The summary and conclusions section ends quite abrupt. Consider to add a short outlook. What are the remaining open questions? Are further field studies planned? You already mention that the effects on ice-clouds is one focus for future endeavors in the motivation.

   b. **(Answer)** We added a new paragraph in the end of the "Summary and conclusions" section, which is reproduced below:

"While the effects of aerosol particles in the warm layer of the clouds is relatively straightforward, this may not be the case for the mixed and frozen portions. An aspect that was not directly addressed in this work is the impacts that warm layer characteristics have on the formation of the mixed phase (above the 0°C isotherm). Given that aerosols alter the properties of the whole warm phase, it is reasonable to assume that this would have an impact on the initial formation of the mixed layer. Such impacts can be in the form of the timing and physical characteristics of the first ice particles and the maximum altitude with supercooled droplets above the freezing level. This issue will be addressed in future studies, taking advantage of data provided by the HALO (High Altitude and Long Range Aircraft) airplane that operated in the second GoAmazon2014/5 IOP between September and October, 2014."

3.

   a. **(Question)** p.4, l.13: Can you tell more about the uncertainties of the instruments? E.g. what is the accuracy of the particle concentrations from the CPC?

   b. **(Answer)** We added the requested information about the accuracy of the instruments. The accuracy for the CPC is ±10%, while for the FCDP it is around 3 µm.

4.

   a. **(Question)** p.6, l.27: You name one factor is commonly cited in literature but do not add any references. I suggest to add some references at this point.

   b. **(Answer)** Added a citation to Abrecht's (1989) work: Albrecht, B.A.: Aerosols, cloud microphysics, and fractional cloudiness. Science 245, 1227–1230, 1989.

5.

   a. **(Question)** p.7, l.18: Calculations show that… -> It would be nice if you shortly can present how you did this estimation.

b. **(Answer)** This affirmative is based on calculations of the averaged second moment in polluted and background clouds. We found that the average second moment for polluted clouds is around twice as the background one. This ratio between the second moment in the polluted/background DSDs is representative of the ratio of the overall surface areas. The text was updated to reflect this change.

6.

a. **(Question)** p.8.,l. 21: While your statement seems to be true for the background clouds, especially for the polluted clouds there seems to be an increase in the last LWC bin. Also the spread is increased. Can the latter be explained by a larger LWC bin size?

b. **(Answer)** Added the sentence at the end of the paragraph: "This effect is clearer in background clouds given the limited aerosol availability". This should make the matter clearer. We believe that the LWC bin size should not have as big of an impact here. Under polluted conditions, new droplets may form even if the LWC is big.

7.

a. **(Question)** Table 2: Add the definition of bottom, mid and top layer to the Table caption.

b. **(Answer)** Added the requested information.

8.

a. **(Question)** Figure 2: This figure looks a bit clumsy. My suggestion would be to create a plot with subfigures with the individual flight plans and add the estimated plume area and the average wind direction for each flight.

b. **(Answer)** We tried several approaches to improve this figure, taking into account the clumsiness and the message we want to get through. The most important thing to show with this figure is that most of the flights had a similar trajectory, which enables the plume classification. The updated figure separates each flight in a subplot, with the plume angular section. We changed the text in order to describe this figure. The last 5 lines of the second paragraph of section 2 is now:

"Figure 2 shows the trajectories for all flights, where the dashed grey lines represent the plume angular section considered from the airplane data. Note that the plume usually disperses from Manaus to the T3 site, with relatively small variations on the direction based on

the wind field. Two flights (4 and 6) had low sampling on the plume given the trajectories and the grey lines may not represent the overall region of the plume. However, the directions identified presented higher CN concentrations than the other ones".

**Phrasing / spelling corrections:**

All phrasing and spelling corrections were addressed. Thank you for taking the time to highlight these issues. The specific corrections you suggested are listed below.

p.1, l.11: in terms *of* aerosol conditions

p.1, l.17: split the sentence after the brackets -> The cloud droplets observed are in the range…

p.1, l.24: correct the superscript of km-1

p.1, l.25: Why you use e.g. for the definition of larger droplets? In my opinion, you can
just omit this.

p.1, l.25: to the cloud base

p.1., l.26: change sentence structure to: The overall shape of the droplet size distribution
(DSD) does not appear to be : : :

p.1, l.31: initiation of the collision-coalescence

p.2, l.4: maintains

p.2, l.8: for the Amazon by Martin et al.

p.2, l.11: that a city like Manaus has on atmospheric conditions

p.2, l.22: Amazonian cloud properties

p.3, l.7: suggests -> suggest

p.3, l.8: over the Manaus area

p.3, l.9: stronger wind component -> dominant wind component

p.3, l.20: clouds microphysical properties -> cloud microphysical properties

p.3, l.25: add comma after background air reference

p.4, l.1: consider to change pollution-aerosols to anthropogenic aerosols

p.4, l.2: are almost only urban, while biomass-burning contribution is very exceptional

p.4, l.6: omit numbered before chronologically

p.4, l.29: what is meant by true airspeed? I guess you mean the speed of the aircraft?

p.6, l.22: by effective size you refer to effective diameter $D_e$?

p.7, l.29: omit *profiles*

p.7, l.30: updraft speeds levels -> updraft speed levels

p.8, l.18: relationships De x LWC and DNC x LWC -> relationships of De and LWC, and of DNC and LWC

p.8, l.20: omit e.g.

p. 9, l.4: omit brackets. Instead write: for each layer, as there are more measurements for lower levels.

p.9, l.14: its mass -> their mass

p.9, l.19: omit e.g.

p.9, l.32: and l.33: once you write plume and once polluted. Try to be consistent.

p.10, l.10: justifies -> explains

p.10, l.10: vertical velocities region -> vertical velocity region

p.10, l.25: the updraft regions DSD -> DSDs in the updraft region

p.11, l.6: Polluted clouds had 10

p.11, l.16: omit e.g.

p.11, l.17: bi-modality *favors* the efficiency

p. 11, l.20: aerosols conditions -> aerosol conditions

Figure 5 caption: affected or not -> affected and unaffected

Figure 5 caption: units of LWC should be $gm^{-3}$

Figure 6 caption: add the information that this is for clouds lower than 1000 m only.

**COMMENTS RECEIVED FROM REVIEWER #2**

**MAJOR CONCERNS**

The manuscript describes unique aircraft measurements of polluted and pristine clouds over the Amazon region during the wet season of 2014. The results are of potential interest not only for those involved in the GoAmazon experiment but to the general ACP audience.

However, the manuscript is poorly written and needs substantial revision to meet ACP standards. Part of the methodology should be better explained and some of the results needs

further investigation. Moreover, I have serious concerns about some methods and the interpretation of results. Hence, I cannot recommend its publication without a major revision.

I have annotated the author's PDF file with many comments and questions to the authors, which I hope will help to improve the manuscript. Here, I only list my major concerns.

**Authors answer**

Dear Reviewer #2, we would like to extend our gratitude for the effort to revise this paper and ultimately helping it be improved. We worked hard to address each one of your concerns individually and this document will detail how we approached it. Following you will find our answers to the major comments you made, while next section will detail the specific comments from the supplement material you provided.

Best regards,

Micael A. Cecchini and co-authors

1.
   a. **(Question)** Introduction needs to be throughly revised. Lines discussing this work are mixed with paragraphs discussing the current state of the art, making it hard to follow for those not part of GoAmazon.
   b. **(Answer)** We revised the introduction with all you recommendations. There are several aspects that were covered by the introduction (i.e. effects of Manaus pollution on air chemistry and cloud physics, differences between wet and dry season, etc...).

2.
   a. **(Question)** Session about the instrumentation should explain what corrections or data processing were performed for the different instruments / probes used. Alternatively, other papers describing that should be cited.
   b. **(Answer)** We updated the text with new information about the instruments – see comments 34-39 below.

3.
   a. **(Question)** Authors used CN to identify if clouds probed under each circunstance were or were not beeing affected by the plume of pollution. I beleive CCN would be better to indicate the influence of the plume on the clouds for 2 reasons. Firstly, because most of the initial pollution particles

emitted will be too small to become CCN, hence the initial plume will not affect much the cloud formation. Secondly, as the plume is chemically and physically transformed downwind of Manaus, the extra aerosols will grow, be oxidized, and thus will interfere more and more with the CCN population. See for instance previous results from Kuhn et al (2010). Hence, as the G1 flight legs are at different distances from Manaus, CCN would be a better indicator than CN.

b. **(Answer)** There are both technical and conceptual reasons why we chose to use CN instead of CCN. The technical one is that there were no CCN measurements for 2 flights, which would make it harder to have a single reference for all flights. Besides, the supersaturations would have to be taken into account, and small differences in supersaturation can add more complexity to our analysis. Our idea was to provide a binary classification, in or out of the plume, with a simple, but strong, criterion adequate to our flights. The conceptual reasons are related to the type of classification we wanted to produce. In this work, we are not analyzing the way the plume changes as it ages and its consequences for cloud formation. The intent is just to locate the plume in order to compile characteristics of clouds growing under its presence and compare to clouds formed under background conditions. The added requirements for the plume classification (i.e. the angular section and the CN concentrations have to be higher than the 90% percentile) further contribute to make sure that the DSD measurements are indeed inside the pollution.

Additionally, from our classification, it is possible to observe that the CCN concentrations are higher for the plume regions even though the classification itself did not take it into account directly. See Figure R.1, for example ("R" refers to revision, so no confusion is made to the paper figures). It shows mean CCN concentrations in different SS for the plume and background regions (for altitudes lower than 1000 m). The CCN concentrations in the plume are more than double the background concentrations for SS=0.23%, while this difference increases with supersaturation. In this way, it is possible to conclude that the plume is able to increase the CCN concentrations and, thus, to affect cloud formation. The second-to-last paragraph of section 2.2 was revised:

"The final result of the classification scheme for March 10 is shown in Figure 4. A visual inspection of radiosonde (released from the Ponta Pelada airport located on southern Manaus)

trajectory plots confirmed the overall direction of the plume for each flight. Given the nature of the meteorology in the Amazonian wet season, i.e. its similarities with oceanic conditions concerning horizontal homogeneity, there should be no significant difference between the thermodynamic conditions inside and outside the plume region for the G-1 flights. In this way, differences observed in pollution-affected clouds are primarily due to the urban aerosol effects. It should be noted that even though the plume classification is defined from the CN measurements, there are also observable differences regarding CCN concentrations. The in-plume CCN concentrations (for altitudes lower than 1000 m) averages at 257 $cm^{-3}$ for a 0.23% supersaturation, while the respective background concentration is 107 $cm^{-3}$ (Figure 5). Note the overall low concentrations, representative of the wet season. In that case, the plume more than doubles the CCN concentrations. For higher supersaturations (which can be achieved in strong updrafts), the differences are even more pronounced. At 0.5% supersaturation, the average CCN concentration inside the plume is 564 $cm^{-3}$, while outside it is 148 $cm^{-3}$. This shows that the plume increases the concentration of aerosol particles that are able to form cloud droplets under reasonable supersaturation conditions, even though they are less efficient than the particles in the background air".

Figure R.1 was added to the manuscript as Figure 5.

[Figure]

**Figure R.1:** CCN concentrations as function of SS. Measurements classified as plume are shown in red, while background is in blue.

4.

    a. **(Question)** For selecting the in-plume events, the authors defined a cone where the plume was most likely to be found. This cone was centered at the airport as if the pollution plume were being dispersed from that single position. While this approach might work for larger distances, for the short distances from Manaus (closer legs) the airport-angle will not confine the plume.

    b. **(Answer)** Yes, this is true and we failed to make it clearer in the text the intention of centering the origin on Manaus airport. By keeping the origin of the coordinate system over the airport, we avoid including measurements over the city on the plume classification because the airport is located on the far west corner of the city. In this way, the heat island effect is avoided, which could introduce different thermodynamic conditions to be considered. See

question 46 in the next section for more details on how we addressed this issue on the manuscript text.

5.

    a. **(Question)** The authors based their whole analyses on a bold hypothesis:

Given the nature of the meteorology in the Amazonian wet season, i.e. (...) horizontal homogeneity, there is no significant difference between the thermodynamic conditions inside and outside the plume region (...). This would be true only for Amazon regions with an uniform vegetation cover, which is not the case at all for the region of Manaus. The Manaus plume goes towards T3, which is the direction of the Solimoes river. Hence, the in-plume cases studied are mostly over or close to the river. On the other hand, the out-plume clouds are far away from the plume and hence from the river. Therefore, as we know from previous studies that the river breeze is significant, one cannot assume that the thermodynamics are the same (over the river and far away)!

To assess the validity of their hypothesis, the authors could, for instance:

- use radiosondes close and away from the river

- look at the specific humidity around the clouds (polluted vs pristine)

- verify the average time of day when polluted vs pristine clouds were sampled

- verify the location (lat/lon) where the polluted vs pristine clouds were sampled

- etc...

    b. **(Answer)** This is a valid concern you expressed and we worked hard to study it and prove that our results really reflect the plume effects and not the river breeze. We found it difficult to prove our point based on the radiosondes, so we will focus on the cloud observations over the rivers or land.

First, it should be noted that most of the flights started at around 10 am local time. At this time the river breeze is not fully developed but is present nonetheless. In this way, there is suppression of the convection over the rivers (Solimões and Negro) and the clouds are predominantly located over land. To check the effects on the DSD measurements, we produced figures similar to

Figure 7 and 8 of the paper substituting the plume-background classification by "over river" or "over land" conditions. Figures R.2 and R.3 shows the averaged DSDs for the vertical layers and w conditions mentioned in the paper. It is quite clear from this that the convection suppression reflects on the growth of droplets. In other words, the droplets tend to be smaller in the clouds over the rivers, with less growth with altitude. This is somewhat expected and the question to address now is how the plume and background classifications relate to the positioning of the rivers.

From the 350 seconds of plume DSDs, 161 are over the rivers and 189 over land. For the background classification, 115 are over rivers and 456 over land. There is a higher contribution of river-DSDs for the plume classification than for the background one. However, the next figures will show that the effects of the pollution on clouds is consistent even considering the land-river differences. Figures R.4 and R.5 shows the averaged DSDs in the same way as in Figure R.2, but restricting for the background and plume classifications, respectively. No DSDs were observed on background conditions over the rivers for the mid and top layers, either by physical or sampling reasons. Focusing on the DSDs over land, it is possible to observe that the plume has a suppression effect on droplet growth, as is seen in Figure 7 of the paper. This shows that the plume effects on the Amazonian clouds is the one noted in the paper even though the clouds over land are more vigorous. We decided to leave the figures unchanged as we feel they consistently represent the effect of Manaus plume on the DSD properties. The manuscript text was updated with comments to this feature in the last paragraph of the methodology section.

[Figure]

**Figure R.2**: averaged DSDs for the bottom, mid and top layers defined in the paper. a) all data, b) only DSDs measured above land and c) only DSDs observed in clouds over the rivers.

[Figure]

**Figure R.3**: the same as Figure 8 in the paper, but the plume DSDs are substituted by those measured over the rivers. The background DSDs are substituted by those observed over land.

[Figure]

**Figure R.4**: the same as Figure R.2, but only considering the points classified as background.

[Figure]

**Figure R.5**: the same as Figure R.2, but only for the points classified as plume.

6.

    a. **(Question)** When authors look at DSD from different altitudes, they divide the vertical from LCL (0%) to freezing level (100%). Then they made averages for relative altitude ranges of 0-20, 20-50 and >50%. There are two things going

on. Firstly, the G1 samples are not well distributed in the vertical, hence the authors had to choose uneven limits to get the same number of samples in each. However, not all shallow clouds will develop as high as the freezing level. Therefore, the average for the bottom layer includes some clouds that did not extend at altitudes >20% and more clouds that did not develop > 50%. On the contrary, samples for the top layer are, by definition, all from clouds that extended from the LCL up to > 50%. Hence, this introduces a large bias. It is mixing clouds of different total vertical development, in different amounts, in each of the three categories. Hence one could not compare DSD from different altitudes, just DSDs from the same altitude for polluted/pristine cases.

Please also note the supplement to this comment:

http://www.atmos-chem-phys-discuss.net/acp-2015-1049/acp-2015-1049-RC2-supplement.pdf

b.  **(Answer)** It is indeed possible that there is a bias regarding cloud top altitude representation in each of our vertical layers. However, we do not believe this should be a concern to our conclusions. Firstly, the main reason to even separate the clouds into the same vertical intervals is to be able to compare different clouds under the same benchmark. We are mainly focusing on comparing plume-affected and background DSDs for the same vertical levels. Otherwise, those layers wouldn't even be needed.

Additionally, there is a common practice by the cloud physicist's community in which measurements made in different clouds in a region can be combined and interpreted as if they were made in a single cloud. For instance, Rosenfeld and Lensky (1998) were able to calculate vertical profiles of effective droplet sizes from satellite based on this assumption. They select a region with clouds with different top heights and use this assumption to produce the profiles as if they were measuring a single well-developed cloud. Freud et al. (2008), using in-situ measurements, found that this assumption is reasonable for the Amazon. We added a comment on the end of the first paragraph of the section "Vertical DSD development and the role of the vertical wind speed":

"Despite probing individual clouds, the DSD measurements can be combined into the three layers defined and interpreted as representative of a single system. It is conceptually similar to satellite retrievals of vertical profiles of

droplets effective radii (e.g. Rosenfeld and Lensky, 1998), where the cloud top radius is measured for different clouds with distinct depths and combined into one profile. This approach was validated with in-situ measurements for the Amazon region by Freud et al (2008)".

**SPECIFIC COMMENTS FROM THE SUPPLEMENT**

The specific comments from the supplement are listed and numbered below. The "q" after the numbers refer to "question" (i.e., the comment from the reviewer), while "a" stands for the answer given by the authors.

**1q)** p.1, l.11: …terms of aerosol concentrations…

**1a)** Done.

**2q)** p.1, l.15: … between the Manaus plume ??

**2a)** Yes, we changed for Manaus plume. The intended meaning is that the city has an effect on the local atmosphere through the production of the plume.

**3q)** p.1, l.17: review the English. You probably want to start a new sentence here. It is also not clear if this range is a result of your observations or an arbitrary cut you choose to use, or a cut imposed by the instruments you had at disposal.

**3a)** We changed the sentence to: "The droplet size distributions reported are in the range 1 μm ≤ D ≤ 50 μm in order to capture the processes leading up to the precipitation formation". In this way the goal is clearer.

**4q)** p.1, l.21: aerodynamic or physical or other? Or doesn't matter for your case?

**4a)** The effective diameter stands for the ratio between the third and second moments of the DSD. In this way, it is a physical diameter and the name of the variable should be self-explanatory.

**5q)** p.1, l.22: the average value has an exact value, hence it cannot range from 10 to 40%. Do you mean "The differences range from 10% to 50%"?

**5a)** We calculated mean effective diameters and droplet number concentrations in each 400 m vertical bins. The "average" was meant to address the vertical averaging process.  However, we left the sentence as you suggested because we feel it sufficiently explains the results without complicating for the reader.

**6q)** p.1, l.22: This is confusing. Of course droplet concentration varies a lot across different vertical levels! But you were talking about differences between polluted and pristine clouds. Please rephrase.

**6a)** We left the final sentence as: "The differences range from 10% to 40% for the effective diameter and are as high as 1000% for droplet concentration for the same vertical levels". We are comparing diameters and concentrations between polluted and background clouds for the same altitudes. Because, as you pointed out, they vary greatly with altitude.

**7q)** p.1, l.30: droplets sizes.

**7a)** Corrected.

**8q)** p.2, l.3: This is not true! The water vapor comes from the ocean!!!

**8a)** Indeed, we removed the "self-contained" to avoid confusion.

**9q)** p.2, l.8: by.

**9a)** Corrected.

**10q)** p.2, l.8: by.

**10a)** Corrected.

**11q)** p.2, l.11: review. You probably want to say "the atmospheric…".

**11a)** We changed "its" for "the", it should be clearer now.

**12q)** p.2, l.12: ,

**12a)** Corrected.

**13q)** p.2,l.12: "around" or "of about".

**13a)** Corrected.

**14q)** p.2, l.13: with

**14a)** Corrected.

**15q)** p.2, l.22: in the case of Manaus.

**15a)** Added.

**16q)** p.2, l.22: focused

**16a)** Corrected.

**17q)** p.2, l.24: study evaluated.

**17a)** Corrected.

**18q)** p.2, l.25: Review the text. Disconnected meanings. You probably want to say something like: "This is important because the city plume is always there, all year long, and polluted clouds…"

**18a)** We changed the sentence to: "However, no study evaluated the urban aerosol interaction with clouds over the rain forest during the wet season, when biomass-burning is strongly reduced given the frequent rain showers that leave the forest wet and more difficult to burn. In this case, the effects of the Manaus plume can be studied separately and in detail. Polluted clouds over the Amazon usually present more numerous but smaller droplets that grow inefficiently by collision-coalescence and therefore delay the onset of precipitation to higher altitudes within clouds (Rosenfeld et al., 2008)".

**19q)** p.2, l.29: Please review. Previous studies showed that precipitation during the monsoon seasons is more stratiform while that during the dry season is more showering. More over, the reason why the wet season is more clean is because the vegetation is wet (not mattering the type of rain) and hence cannot be burned.

**19a)** Regarding the cleaner atmosphere for the period, we rephrased in order to make clearer that the air is clean due to the reduction in biomass burning. The sentence is now: "The period is in the wet season, which presents a clean atmosphere due to the reduction in biomass burning". In the previous question, we included one comment saying that there is less biomass burning because the forest is wet.

The clouds are surely more convective during the dry season. However, cumulus fields are very common during the wet season, being frequent during the campaign days. We observed stratiform rain during the campaign, but we can confirm that most of the clouds during the flights were cumulus. We are not sure what to review in this sentence related to this discussion given that we don't mention the convective x stratiform regimes here.

**20q)** p.2, l.31: It is not the background air who provides the opportunity. Please rephrase.

**20a)** The sentence is now: "The pristine characteristic of the background air provides the opportunity for contrasting the microphysics of natural and urban pollution-affected clouds".

**21q)** p.2, l.31: Unless you are going to state that this is the focus of GoAmazon, I don't see why you should talk about chemistry if you will focus on clouds.

**21a)** See previous question.

**22q)** p.3, l.2: you probably mean something else.

**22a)** The sentence is now: "This scenario allows for the first time the direct comparison between clouds formed under background conditions and those affected by pollution in the wet season".

**23q)** p.3, l.3: you probably mean clouds formed under background conditions.

**23a)** see previous question.

**24q)** p.3, l.4: you will only analyze data from the wet season. It is not clear then why you are discussing difference between the dry/wet seasons. You didn't even mention what was found by Machado (2004) in terms of cloud microphysics.

**24a)** The intent of this paragraph is to show that there is a large scale forcing for the clouds during the wet season. This forcing is related to the monsoon system. We feel that it is important to give a general picture of the large scale features in place during the campaign. The paragraph mentions that the large scale is related to the monsoon system and further details can be seen in the citations given.

**25q)** p.3, l.18: give the number or rephrase.

**25a)** Gave the number – 16.

**26q)** p.3, l.26: see my comment on figure 1. It is very confusing!

**26a)**

**27q)** p.3, l.29: past tense.

**27a)** Corrected.

**28q)** p.3, l.31: past tense.

**28a)** Corrected.

**29q)** p.3, l.31: past tense.

**29a)** Corrected.

**30q)** p.4, l.3: sections.

**30a)** Corrected.

**31q)** p.4, l.4: past tense.

**31a)** Corrected.

**32q)** p.4, l.4: past tense.

**32a)** No correction needed as far as we know.

**33q)** p.4, l.7: this should be at the beginning of the section.

**33a)** Relocated the sentence to the beginning of the section. It is now the second sentence of the Methodology section.

**34q)** p.4, l.11: Typical CPC that use butanol measure only > 10nm. Did the G1 had an ultra-fine CPC? If that is the case, you should say. Moreover, what are the losses on the G1 inlet? Does it allow particles of 3nm or 3 microns to reach the CPC?

**34a)** The characteristics of the CPC used are the ones given in the text. The cut-off diameter is 3 nm, making it sensible to small particles and, therefore, able to readily detect urban pollution. The intent of using this CPC instead of the model 3010 (also available on the plane) is to better locate the plume, even though particles as small as 3 nm are not able to activate droplets. As mentioned in the text, the intent of using CN concentrations instead of CCN is to produce a qualitative classification of the plume. Note that the quantitative CN values are not used in any part of the study. We consider that there is not enough statistics to analyze DSD characteristics for several levels of pollution. That is one of the reasons we only produce a binary classification, it is a way to characterize clouds inside and outside the plume regardless of the exact amount of CCN produced by the urban pollution.

As mentioned in question 35 (below), inlet losses are lower than 4% (penetration higher than 96%), with an up limit of 5 µm in diameter.

**35q)** p.4, l.13: Please give a reference. TSI instruments, for instance, measure only up to 10000. After that point, the chance of coincidence is non-negligible.

**35a)** We updated the manuscript text with more information on the CPC instrument used (model 3025). The first paragraph of Section 2.1 is now:
"The two main instruments used for this study were the Condensational Particle Counter (CPC, TSI model 3025), and the Fast Cloud Droplet Probe (SPEC Inc., FCDP). The CPC instrument

measures number concentration of aerosols between 3 nm and 3 μm using an optical detector after a supersaturated vapor condenses onto the particles, growing them into larger droplets. Particle concentrations can be detected between 0 and 105 cm-3, with an accuracy of ±10%. Coincidence is less than 2% at 104 cm-3 concentration, and corrections are automatically applied for concentrations between 104 cm-3 and 105 cm-3. The CPC was mounted in a rack inside the cabin and connected to an isokinetic inlet and an aerosol flow diluter and was operated using an external pump. The isokinetic inlet has an up limit of 5 μm for particle diameter, with penetration efficiency higher than 96%. A 1.5 LPM flow rate was maintained using a critical orifice. The dilution factor varied between 1 and 5".

**36q)** p.4, l.19: aircraft measurements are tough! You should be clear about which corrections were applied or not to the data, how it was cleaned, etc…

**36a)** We added sentences explaining the filter applied to the FCDP-measured DSDs. The end of the second paragraph in Section 2.1 now is as follows: "Shattering effects were filtered from the FCDP-measured DSDs (Droplet Size Distributions), which is a built-in feature of the provider software. Additionally, measurements with low number concentrations (< 0.3 cm$^{-3}$) and low water contents (< 0.02 gm$^{-1}$) were excluded".

For the CPC measurements, we changed the first paragraph of Section 2.1 – see question 35 above.

**37q)** p.4, l.21: how many % of the dataset had to be interpolated? You should also mention how you average the data, as I don't think you used 1Hz… If you average over 30s or more, I don't see why you would need this interpolation.

**37a)** Around 16% of the CPC data was flagged as "bad" and was interpolated, while 0.02% was excluded because no good measurement was close enough. Additionally, we performed tests with moving averages on the CN data and noted no significant impact on the results with periods of up to 10 s. Higher averaging periods seem too large, given that the airplane flew at around 100 ms$^{-1}$ and 10 s represents roughly 1 km. In this case, we chose to continue using 1 Hz measurements. The second paragraph of Section 2.1 was changed in order to reflect those comments:

"The quality flag of the CPC instrument was used to correct the concentration measurements. Whenever an observation was flagged as "bad", it was substituted by an interpolation between the closest measurements before and after it that were either "questionable" or "good". For "good" measurements, which represents 59% of all the measurements, the

uncertainty is less than 10%. The interpolation weights decayed exponentially with the time difference between the current observation and the reference ones. If the reference observations were more than 10 s apart, these data were excluded. 16% of the data was interpolated in that manner, while only 0.02% had to be excluded. This process was required not only to smooth out the bad measurements but also was important to maintain significant sample sizes (instead of simply excluding "bad" measurements). No averaging was applied to the 1 Hz CPC data. However, tests were made in order to check the impact that the sample frequency had on the results. The results were not sensible to moving averages of up to 10 seconds, which corresponds to roughly 1 km displacement given that the G-1 flew around 100 ms$^{-1}$ in speed. Given this observation, the analyzes are based on the 1 Hz CPC measurements".

**38q)** p.4, l.29: do you mean accuracy (distance from the true value) or precision distance from average value)? If you mean accuracy, how was it even calculated? Which other instrument has been used for giving the true vertical wind speed?

**38a)** We mean precision, the text is corrected.

**39q)** p.4, l.29: was 75m/s the approximate G1 speed? The accuracy of 0.75m/s is too large when compared to the typical vertical wind speeds (1-3m/s). You should try to estimate how much miss-classifications you might have.

**39a)** The updated text after question 37 notes that the G-1 speed is around 100 ms$^{-1}$. Therefore, the vertical wind speed precision should be close to 0.75 ms$^{-1}$. This is one of the reasons we use relatively wide w bins in Figure 8. We believe there is not much impact of miss-classifications in that case.

**40q)** p.5, l.2: these are too small to become CCN.

**40a)** They may be right after emission, but they become more efficient CCN as they age. We updated the sentence to reflect this: "Urban activities such as traffic emit large quantities of particles to the atmosphere, which are then transported by atmospheric motions and can participate in cloud formation, especially when they grow, age and become more effective droplet activators".

**41q)** p.5, l.4: it will only affect if they have the size (and chemistry) to compete for the available water vapor.

**41a)** The urban aerosol are indeed smaller than the background ones and are less effective to become CCN. However, their high concentrations are able to produce more CCN even so. See, for instance, Fig. 4 in Kuhn et al (2010) – the study you mention in the next question.

**42q)** P.5, l.6: why not using CCN? Was it not measured by G1? As the plume is chemically and physically transformed downwind of Manaus, the extra aerosol loading will interfere more and more with the CCN population. See for instance Kuhn et al (2010). Hence, as the G1 flight legs are at different distances from Manaus, CCN would be a better indicator than CN.

**42a)** See major question 3 in the previous section.

**43q)** p.5, l.19: it is not clear what you mean.

**43a)** We believe this is explained in the next sentence on the manuscript (i.e. a CN measurement inside the cloud is substituted by the closest cloud-free measurement).

**44q)** p.5, l.20: You are using CN just to build a plume/background mask… so why making it so complicated? For instance, what happens if CN before the cloud says "plume" while CN after the cloud says "background"? By your criteria of time-distance, half of the plume will be polluted and half will be clean. Does it make sense? Why don't you do the mask on a cloud basis instead of 1Hz basis?

**44a)** As mentioned before in this document and also on the manuscript, most of the clouds probed were part of the Cu fields usually observed during the wet season. This makes it almost impossible to have a "background" classification in one side and a "plume" classification on the other side of the cloud because of the size of the systems. Additionally, the 90% percentile requirement on the CN concentrations for the plume classification results in measurements closer to the center of the plume. The "background" classification also requires the measurements to be outside the plume angular section in order to avoid this issue.

**45q)** p.5, l.26: so you are throwing out 65% of the data?

**45a)** Not necessarily, and the percentage is actually 74%. There are CPC measurements not only during the cloud penetrations but also in clear sky. The 90% and 25% percentiles refer only to the CPC measurements. Only a portion of the CPC data points have a corresponding FCDP one. Even though most of the data remains unclassified, we believe it is necessary in order to differentiate as much as possible both populations.

You may have noted that the number of measurements changed slightly in this new version of the manuscript (see page 6, line 12). We changed the DSD filtering slightly, eliminating the

cases where DNC < 0.3 cm$^{-3}$ and LWC < 0.02 gm$^{-3}$. Before we only eliminated the DSDs where LWC < 0.02 gm$^{-3}$. The change is to make it more consistent with the cloud flag we use – described in the methodology. Minimal impacts on the dataset resulted from this and no impact whatsoever on the results.

**46q)** p.6, l.3: This is not true. The plume does not originate on the airport!! Hence, calculating theta from there is misleading, particularly for the short distances from Manaus. See, for example, the dark blue lines on Fig. 3. The plume is much wider than what is indicated by the vertical lines, exactly because of this reason!

**46a)** Yes, this is true and we failed to make it clearer in the text the intention of centering the origin on Manaus airport. By keeping the origin of the coordinate system over the airport, we avoid including measurements over the city on the plume classification because the airport is located on the far west corner of the city. In this way, the heat island effect is avoided, which could introduce different thermodynamic conditions to be considered. The manuscript text was updated in the following manner, starting from the line of the comment:

"Note that there is an angular section where the concentrations are high not only close to the city but also as far as 70 km. This section is defined to be affected by Manaus pollution plume (delimited by grey dashed lines in Figure 3). Note that the coordinate system is centered on Manaus' airport, where the G-1 took off, and not on the center of the city or other point of interest. For this reason, it is also possible to observe relatively high CN concentrations close to the origin and to the northeast and southeast directions. This corresponds to high CN concentrations over the city. By keeping those directions outside the plume angular section, this data is not considered as plume. This is intentional because other aspects occur over the city that may contribute to the cloud formation. For instance, the heat island effect may contribute to the convection, changing the thermodynamic conditions compared to those over the forest. By keeping the origin point as the airport, which is located on the west section of the city, this problem is avoided."

**47q)** p.6, l.5: where is this plot? And why didn't you look at the radio sonde from the ponta-pelada airport? It makes much more sense, as it will travel between Manaus and T3.

**47a)** We did not feel the need to show this plot, we are just confirming that we looked into it. We feel that it is quite reasonable that the plume and the radiosonde would have a similar trajectory in lower level given that they are subject to the same wind field. The mentioned radiosondes were not released from T3, it was a mistake on the text. They were indeed released from Ponta Pelada airport, thanks for pointing that out.

**48q)** p.6, l.8: this would be true only if considering an uniform vegetation at the surface. Which is not the case at all for the region of Manaus. The Manaus plume goes towards T3, which is the direction of the Solimões river. Hence, the in-plume cases you selected are, I guess, mostly over or close to the river. On the other hand, the out-plume clouds will be far away from the plume and hence from the river. Therefore, you cannot assume waving hands that the thermodynamics are the same! But you have the radiosondes from T0z and from T3. You should compare them to prove.

**48a)** This question is addressed in the item 5 of the major concerns (previous section in this document). Please refer to it.

**49q)** p.6, l.11: this is not the first place CCN is defined.

**49a)** Corrected.

**50q)** p.6, l.19: what about the environmental specific humidity outside the clouds? As I said, probably the environmental conditions are not the same for the clean and plume samples.

**50a)** The wet season in Manaus and its surroundings is characterized by very high relative humidity values (e.g.  90%) given the constant inflow of humidity from the trade winds. We believe the humidity outside the clouds should be similar for all cases.

**51q)** p.6, l.25: that only means that the difference in LWC is smaller than the difference in DNC.

**51a)** Changed the sentence to: "This factor shows that, despite condensing lesser amounts of total liquid water, the background clouds are able to produce bigger droplets than their polluted counterparts". The issue should be clearer now.

**52q)** p.6, l.28: Please note that availability of water is different than LWC. Hence you should look at the specific humidity around the polluted and pristine clouds (or below their cloud bases) to be able to say that the water availability is different… or that aerosols have an effect on that.

**52a)** The idea is to separate the bulk condensation efficiency of the clouds (that affects LWC values) and the water vapor competition scenario (which is usually analyzed in fixed LWC). The sentence is now: "The other factor is how much bulk water the systems are able to condense while the vapor competition is ongoing".

**53q)** p.7, l.4: under polluted conditions?

**53a)** Yes, corrected.

**54q)** p.7, l.4: slower? If droplets are smaller, the condensation ratio is lower, and hence they grow slower then initially larger droplets, isn't it? You even say that later on…

**54a)** The smaller droplets in polluted conditions grow faster by the condensation process only – the condensation rate is inversely proportional to size. The droplets under background conditions grow faster overall, because they anticipate the start of the collision-coalescence process.

**55q)** p.7, l.13: But that would change the LWC near the top of the cloud, where droplets could be large enough to precipitate. This, by the way, rings a beçç: it does not make sense to include in Fig. 5 data from all altitudes. At this point you are discussing the impact of aerosols on the droplet formation at cloud base… Hence the analysis would be more coherent if Fig. 5 showed on near-base data. If you do for both (base x top) you may be even able to disentangle the two mechanisms you identified.

**55a)** By comparing the histograms in the different layers of the clouds, the same observations are possible (i.e. higher LWC and NDC and lower De for polluted clouds). Se Figure R.6 and R.7 below. In this way, it is possible to observe that the aerosols affected the whole warm layer structure of the clouds. We chose to leave the manuscript unchanged in that regards as it already illustrates the issue adequately.

[Figure]

**Figure R.6:** the same as Figure 5 in the paper, but only for the bottom layer.

[Figure]

**Figure R.7:** the same as Figure 5 in the paper, but only for the mid and top layers.

**56q)** p.7, l.17: hence hypothesis 2 is bad.

**56a)** We believe hypothesis 1 should be the most significant, therefore we added a sentence right after: "The second process identified (i.e. suppressed precipitation staying longer inside the clouds) probably has a lesser impact".

**57q)** p.7, l.19: why would that be? You are talking about the area of the droplets... and then go back to a previous step to consider that what more CCN could do? Isn't there a confusion between the aerosol area and the droplet area? I mean, when you first discussed hypothesis #1 you are talking about the larger area of aerosol surface under polluted conditions... But then you calculated the area for your DSD.

**57a)** Although initially the vapor condenses onto the aerosol particles, when the droplets form the vapor continues to condense onto them. By calculating the average second moment of the polluted and background DSDs, it is possible to calculate the overall area available for condensation onto the droplets. In this way, we do not need to look into aerosol size distributions and can focus solely on the cloud-DSDs.

**58q)** p.7, l.20: We have already enough evidence (at least published on conferences) to show that under polluted conditions (urban or biomass burning) the organic (75%)/inorganic (25%) fractions are the same. Besides, we also have hygroscopicity measurements showing lower values that under polluted conditions because of the much lower hygroscopicity of the organic component.

**58a)** Indeed, we removed the hygroscopicity mention.

**59q)** p.7, l.27: why would it not be?

**59a)** Correct, we changed "considering that…" to "given that…".

**60q)** p.7, l.31: (hopefully) the non-precipitating shallow cumulus. But why < 1km? You should restrict yourself to the cloud base, and I find 1km too deep for shallow clouds. You should justify why 1km and not 300m or 2km.

**60a)** There are two reasons for choosing 1km instead of 500m or 300m. Firstly, the amount of data more than doubles from 500m to 1km. Additionally, in polluted clouds the aerosol activation process may last longer and higher in the cloud when compared to the background clouds. For that reason, the higher LWC bins can be underrepresented with a 500m limit. We added a sentence following your comment: "The 1000m limit is chosen for both maximizing statistics and also capturing the layer in which the aerosol activation takes place. That layer is possibly thicker under polluted conditions, given the higher availability of nuclei".

**61q)** p.7, l.34: it is influenced by the aerosol population with sizes allowing it to be activated. But, since you have aerosol size distribution and CCN measured on the G1… Why don't you estimate LWC for each updraft (ie. For each max SS)?

**61a)** It is not the intent of this paper to model the expected LWC from the aerosol size distribution. We defined a strategy to locate the plume and the background regions and accumulate statistics for each case. What this affirmation means is that the clouds affected by the plume presented higher LWC, which can only be associated to aerosols given we eliminated thermodynamic conditions. In Kuhn et al. (2010) paper it is possible to see that CCN is enhanced in Manaus plume, showing that the pollution increases the number of aerosol particles that have sizes above the critical diameter to activate. As we mentioned before, the intent here is not to analyze the DSD properties for specific quantitative CCN values. Instead, we report on the mean microphysical properties of clouds formed in and out the plume.

**62q)** p.8, l.1: Come on, you cannot say that by just visual inspection of the red points in the figure 6a! You have to make a line fit (considering the error bars!!) and make a null hypothesis test. Hence setting a confidence level for your statement.

**62a)** We agree that a more robust statistical analysis would be required to estimate the confidence level. However, the physical processes identified are consistent to the average results we obtained. As such, we left the confidence level question open (a bigger dataset is desirable), but discussed the possible physical mechanisms. The paragraph is now (starting from the sentence of this question):

"This figure shows that, on average, not only are the polluted clouds more efficient at the bulk water condensation but also the resulting LWC scales with updraft speed (linear coefficients, considering the error bars, are 0.13 g s m$^{-2}$ for plume measurements and 0.033 g s m$^{-2}$ for background clouds). In a background atmosphere, most of the aerosols readily activate, and increases in updraft strength does not result in further condensation. On the other hand, the higher availability of aerosols inside the plume allows for more condensational growth as long as enough supersaturation is generated, especially considering that the critical dry diameter for activation is inversely proportional to supersaturation and, consequently, to the updraft speed. However, a deeper analysis in a bigger dataset would be required to assess the statistical significance. The enhanced condensation efficiency and the possible LWC scaling with updraft strength at least partly explain the higher liquid water contents in the plume-affected clouds. The standard deviation bars in Figure 6a indicate that while there is high variability for the LWC in polluted clouds, the clean ones are rather consistent regarding the condensation efficiency".

**63q)** p.8, l.12: again, you have to do an statistical test.

**63a)** Changed the sentence to: "It is clear that, even with the dispersion observed, the two DSD populations present consistently different average behaviors for all LWC intervals".

**64q)** p.8, l.19: why almost? Please make the line fit and statistical tests, so that you can state that with statistical confidence.

**64a)** Removed the "almost" and included R$^2$ information to show that we calculated the linear fits.

**65q)** p.8, l.22: by your own argument, this might not be the case. You can have a low LWC content at higher altitude inside the cloud that did not develop high LWC. I understand that

you are onlu showing data < 1000m from cloud base, but this might be too deep. It is not obvious that  low LWC means cloud base high LWC means near 1km.

**65a)** It is true that LWC usually increases with altitude both for polluted or background clouds. It even has a more pronounced increase in clouds under the effect of the plume. Given that LWC increases with altitude, this factor is implicit in our affirmation. We said that for low LWC (or closer to cloud base), increases in DNC have a higher impact on the LWC value. However, for higher LWC (or higher in the cloud), new droplet formation won't affect as much LWC because the cloud droplet are bigger – LWC depends cubically on the diameter.

**66q)** p.8, l.22: at this stage of the cloud life you are discussing, the droplets present are those activated from aerosols at cloud base. The number of droplets activated on the cloud base depend (among other things) on the maximum SS achieved (i.e updraft). Hence, it might make sense to plot DNC x W near cloud base…

**66a)** It is indeed an interesting analysis. Se Figure R.8 below with this calculation (graph b). It shows that, on average, DNC is always higher for plume-affected clouds and it tends to grow with w. However, we think that the discussion presented on the paper already covers the main physical mechanisms at play and this graph is not actually needed, despite being interesting.

[Figure]

**Figure R.8:** the same as Figure 5 in the paper, but adding averaged DNC according to w intervals (b).

**67q)** p.9, l.2: There are two things at the same time going on. Firstly, the G1 samples are not well distributed in the vertical, hence you had to choose uneven limits to get the same number of samples. However, another point is that not all shallow clouds develop as high as the freezing level (yours 100%). Hence, your average for the bottom layer includes some clouds that did not extend at altitudes > 20% and more clouds that did not develop > 50%. On the contrary, your samples for the top layer are, by definition, all from clouds that extended from the LCL up to > 50%. You have, thus, and important bias! You are mixing clouds of different total vertical development, in different amount, in each of the three categories.

**67a)** It is indeed possible that there is a bias regarding cloud top altitude representation in each of our vertical layers. However, we do not believe this should be a concern to our

conclusions. Firstly, the main reason to even separate the clouds into the same vertical intervals is to be able to compare different clouds under the same benchmark. We are mainly focusing on comparing plume-affected and background DSDs for the same vertical levels. Otherwise those layers wouldn't even be needed.

Additionally, there is a common practice by the cloud physicist's community in which measurements made in different clouds in a region can be combined and interpreted as if they were made in a single cloud. For instance, Rosenfeld and Lensky (1998) were able to calculate vertical profiles of effective droplet sizes from satellite based on this assumption. They select a region with clouds with different top heights and use this assumption to produce the profiles as if they were measuring a single well-developed cloud. Freud et al. (2008), using in-situ measurements, found that this assumption is reasonable for the Amazon.

**68q)** p.9, l.14: check the quartiles. There seems to be an error as they, sometime, go to zero at the begin/end of each curve. They also, in some case, do not contain the average value.

**68a)** When the quartiles go to zero it only mean that 25% or 75% of the respective bins concentrations are 0. For instance, the polluted DSDs on the bottom layer present, in most cases (higher than 75%), bin concentrations equal to 0 for D>20 μm. In that case, the quartiles will go to zero in that size range.

The average is not required to be inside the interquartile range, do not take it by the median. When the average is outside the interquartile range, it means that a few measurements presented high enough concentrations to bring the mean value even higher than the 75% quartile. When the average is not null and the quartiles are zero, it means that there are only a few DSDs (frequency lower than 25%) contributing to the respective size range.

**69q)** p.10, l.8: please check the quartiles and averages, as in the last figure.

**69a)** See previous question.

**70q)** p.10, l.17: the size of the activated droplet depend mostly on the super saturation and not on the aerosol.

**70a)** The size of the aerosol where the water is condensing defines the initial size of the droplet. As such, bigger aerosols would favor the formation of bigger droplets. If this was not the case, giant CCNs would have a similar impacts on the clouds as a smaller CCNs.

**71q)** p.12, l.22: Please review all references. This one, for example, was not cited in the text.

**71a)** We cited Kuhn et al. (2010) while answering question 41. Checked all other citations to make sure everything is correct.

**72q)** p.19, l.2: what is theta?

**72a)** Changed the sentence to: "θ is the azimuth angle and is zero for East direction and grows counterclockwise". Theta is the azimuth angle.

**73q)** p.20, l.1: these fluctuations at higher altitudes doesn't make sense… it clearly shows that you have lower statistics and hence large standard deviation. You should decrease the vertical resolution.

**73a)** Changed the resolution to 800m, no significant impacts on the results.

**74q)** p.22, l.4: please make it centered. You choose a log-x scale, hence it is not possible to infer the limits (and hence the center) of each horizontal bin.

**74a)** It is centered now.

**75q)** p.23, l.1: It seems there is something wrong. The average value (dark blue) is not within the 25-75% quantile (light blue). Interquartile range for red and green goes to zero at larger sizes.

**75a)** See question 68.

**MARKED-UP MANUSCRIPT
STARTS ON NEXT PAGE**

[revised manuscript text omitted]